# Vacuolar H⁺-ATPase determines daughter cell fates through asymmetric segregation of the nucleosome remodeling and deacetylase complex

**Zhongyun Xie[1], Yongping Chai[1]\*, Zhiwen Zhu[1], Zijie Shen[1], Zhengyang Guo[1], Zhiguang Zhao[2], Long Xiao[2], Zhuo Du[2], Guangshuo Ou[1], Wei Li[3]\***

[1]Tsinghua-Peking Center for Life Sciences, Beijing Frontier Research Center for Biological Structure, McGovern Institute for Brain Research, State Key Laboratory of Membrane Biology, School of Life Sciences and MOE Key Laboratory for Protein Science, Tsinghua University, Beijing, China; [2]State Key Laboratory of Molecular Developmental Biology, Institute of Genetics and Developmental Biology, Chinese Academy of Sciences, University of Chinese Academy of Sciences, Beijing, China; [3]School of Medicine, Tsinghua University, Beijing, China

**\*For correspondence:**
ypc@mail.tsinghua.edu.cn (YC);
weili_med@mail.tsinghua.edu.cn (WL)

**Competing interest:** The authors declare that no competing interests exist.

**Abstract** Asymmetric cell divisions (ACDs) generate two daughter cells with identical genetic information but distinct cell fates through epigenetic mechanisms. However, the process of partitioning different epigenetic information into daughter cells remains unclear. Here, we demonstrate that the nucleosome remodeling and deacetylase (NuRD) complex is asymmetrically segregated into the surviving daughter cell rather than the apoptotic one during ACDs in *Caenorhabditis elegans*. The absence of NuRD triggers apoptosis via the EGL-1-CED-9-CED-4-CED-3 pathway, while an ectopic gain of NuRD enables apoptotic daughter cells to survive. We identify the vacuolar H⁺–adenosine triphosphatase (V-ATPase) complex as a crucial regulator of NuRD's asymmetric segregation. V-ATPase interacts with NuRD and is asymmetrically segregated into the surviving daughter cell. Inhibition of V-ATPase disrupts cytosolic pH asymmetry and NuRD asymmetry. We suggest that asymmetric segregation of V-ATPase may cause distinct acidification levels in the two daughter cells, enabling asymmetric epigenetic inheritance that specifies their respective life-versus-death fates.

## eLife assessment

The authors make the intriguing proposal that the NuRD complex in *C. elegans*, which has been linked to the regulation of the cell death protein EGL-1 before, becomes asymmetrically distributed after cell division and that this asymmetry relies on V-ATPase activity. Whereas some disagreement remained between the reviewers' and the authors' interpretation, the final version incorporated alternative possibilities in the text, and with careful interpretation, the current article's model is supported by **solid** data, and represents a **valuable** contribution to the field.

## Introduction

Asymmetric cell division (ACD) gives rise to two daughter cells that possess identical genetic material but distinct cell fates, playing a crucial role in both development and tissue homeostasis (*Royall and Jessberger, 2021*; *Sunchu and Cabernard, 2020*; *Venkei and Yamashita, 2018*; *Wooten et al., 2020*; *Zion et al., 2020*). Both extrinsic and intrinsic mechanisms determine distinct daughter cell

fates after ACD. While extrinsic mechanisms, such as exposure to signaling cues from the local niche, have been extensively studied and are known to define stem cell fate (*Morrison and Spradling, 2008*), the intrinsic mechanisms are more complex and largely unresolved, despite several proteins, RNA molecules, and organelles having been implicated in the regulation of some types of ACD (*Sunchu and Cabernard, 2020*; *Zion et al., 2020*). Epigenetic mechanisms play a crucial role in guiding the two daughter cells toward establishing differential gene expression profiles, ultimately defining their unique cell identities (*Allis and Jenuwein, 2016*; *Escobar et al., 2021*; *Stewart-Morgan et al., 2020*). During the ACD of *Drosophila* male germline stem cells (GSCs), pre-existing and newly synthesized histones H3 and H4 are asymmetrically segregated toward the stem daughter cell and the differentiating daughter cell, respectively (*Wooten et al., 2020*). Nevertheless, the extent to which epigenetic information is asymmetrically inherited through ACD in other organisms and the mechanism by which this process occurs remains elusive.

*Caenorhabditis elegans* represents a valuable model for investigating ACD, given its invariant cell lineage and conserved mechanisms of ACD. During hermaphrodite development in *C. elegans*, 131 somatic cells undergo programmed cell death, with the majority produced through ACD that create a large cell programmed for survival and a small cell programmed to die (*Sulston and Horvitz, 1977*; *Sulston et al., 1983*) The opposing cell fates of daughter cells, that is, to live or die, offer a compelling experimental system for investigating how epigenetic inheritance determines life-versus-death decisions during ACD. It is noteworthy that 105 of the 131 apoptotic cells arise from neuronal lineages, of which the Q neuroblast represents a tractable system for studying ACD and apoptosis at single-cell resolution (*Ellis and Horvitz, 1986*; *Hedgecock et al., 1983*; *Ou et al., 2010*; *Sulston and Horvitz, 1977*). During the first larval stage, Q neuroblasts on the left (QL) and right (QR) undergo three rounds of asymmetric divisions, which produce three different neurons and two apoptotic cells (Q.aa and Q.pp), respectively (*Figure 1A*).

In the nematode, the classic apoptotic pathway initiates upon activation of the BH3-only protein EGL-1 in the cells that are fated to die. EGL-1 binds to the anti-apoptotic Bcl-2-like protein CED-9, which facilitates the release of the pro-apoptotic protein CED-4, leading to caspase CED-3 activation. Activated caspase promotes the exposure of phosphatidylserine (PS) on the surface of apoptotic cells, which triggers the phagocytosis of apoptotic cells via partially redundant signaling pathways such as the CED-1/CED-6/CED-7 pathway and the CED-2/CED-5/CED-12 pathway. Proper transcriptional regulation of the *egl-1* gene is vital for life-versus-death decisions in *C. elegans* (*Conradt and Horvitz, 1998*; *Nehme and Conradt, 2008*). Although the cell-specific transcriptional regulator has been identified in certain cells, including the sexually dimorphic HSN and CEM neurons, M4 motor neuron, NSM neuron, P11.aaap cell, ABpl/rpppapp, and the tail spike cell (*Hirose et al., 2010*; *Jiang and Wu, 2014*; *Nehme and Conradt, 2008*), the general upstream regulators of *egl-1* at the chromatin remodeling or epigenetic level are unclear. Two NSM neuroblasts undergo ACDs to each generate a larger NSM cell programmed to survive and a smaller NSM sister cell programmed to die (*Sulston et al., 1983*). The PIG-1 kinase-dependent asymmetric partitioning of the Snail-like transcription factor, CES-1, represses *egl-1* transcription in the larger NSM sister cells, thereby preventing their apoptosis (*Hatzold and Conradt, 2008*; *Wei et al., 2020*) However, the mechanism underlying the asymmetrical activation of *egl-1* in other ACDs remains elusive.

In this study, we demonstrate the enrichment of the NuRD complex in cells that are predetermined to survive, and its role in suppressing the EGL-1-CED-9-CED-4-CED-3 apoptotic pathway through repression of the *egl-1* gene. Furthermore, we report the interaction between the NuRD complex and the V-ATPase complex, and reveal that during ACD, the localization of the V-ATPase complex and cytoplasmic pH are asymmetrical, thereby contributing to the polarized segregation of the NuRD complex.

## Results

### NuRD asymmetric segregation during neuroblast ACDs

In order to gain insights into the molecular distinctions between apoptotic and surviving cells in *C. elegans* embryos, we employed the SPLiT single-cell RNA sequencing (scRNA-seq) platform to compare their transcriptomes (*Figure 1—figure supplement 1A–C*; *Rosenberg et al., 2018*). The expression of the critical somatic apoptosis-inducing gene *egl-1* was used to distinguish the apoptotic

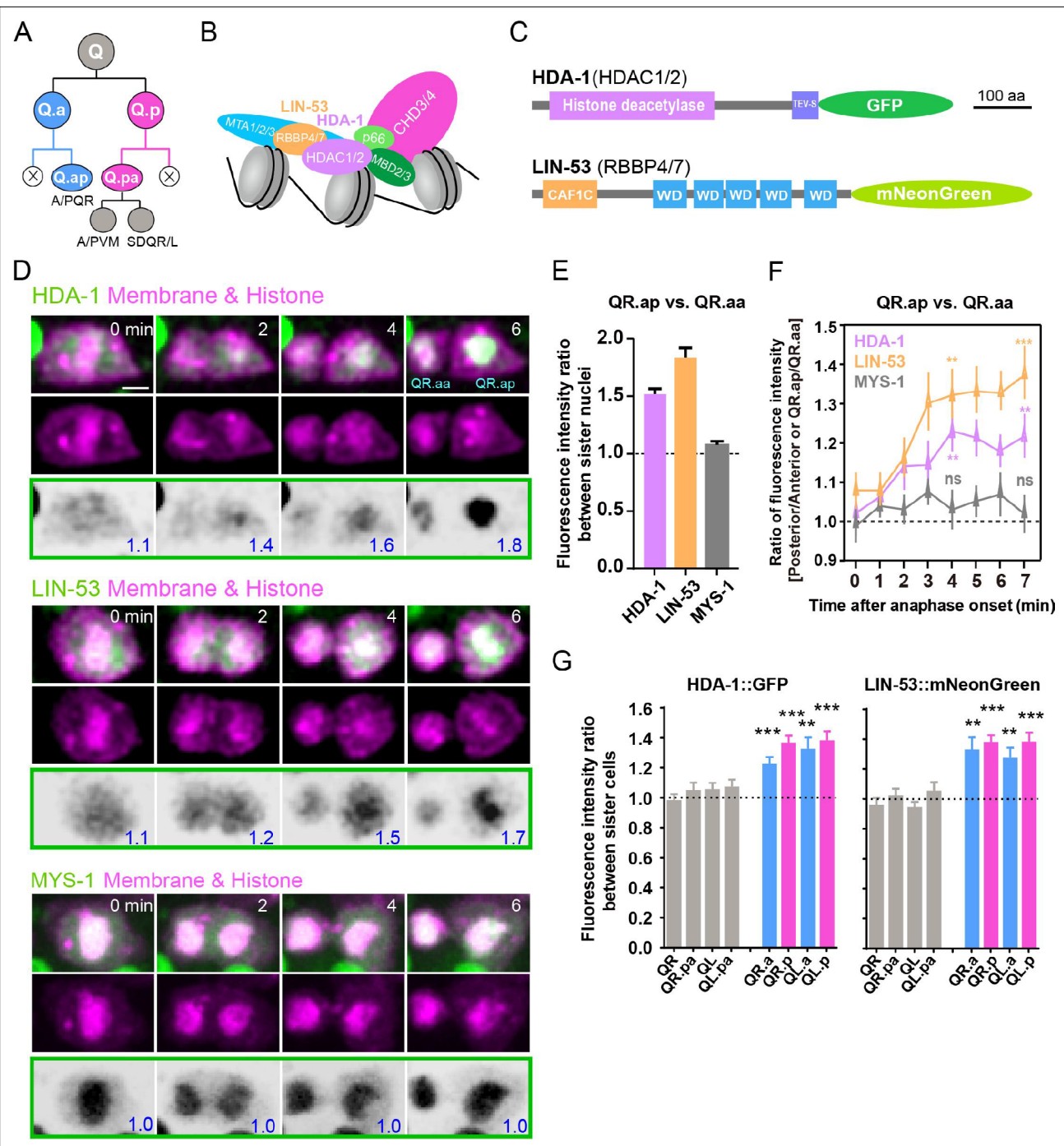

**Figure 1.** Asymmetric segregation of nucleosome remodeling and deacetylase (NuRD) during asymmetric cell divisions (ACDs) of *C. elegans* Q neuroblast. (**A**) Schematic of the Q neuroblast lineages. QL or QR neuroblast each generates three neurons and two apoptotic cells (Q.aa/Q.pp, X). QL produces PQR, PVM, and SDQL. QR produces AQR, AVM, and SDQR. (**B**) A model of the NuRD complex composition (*Bracken et al., 2019*; *Lai and Wade, 2011*). (**C**) Protein domain structure of the GFP-tagged HDAC1/2 (HDA-1) or mNeonGreen-tagged RBBP4/7 (LIN-53). CAF1C: histone-binding protein RBBP4 or subunit C of CAF1 complex; WD: WD40 repeat tryptophan-aspartate domain. Scale bar: 100 amino acids. (**D**) Representative images of endogenous HDA-1::GFP, LIN-53::mNeonGreen and overexpressed MYS-1::GFP during ACDs of QR.a. In each panel, the top row shows merged images, the middle row shows mCherry-tagged plasma membrane and histone, and the bottom row shows inverted fluorescence images of GFP/ mNeonGreen. The anterior of the cell is on the left. The GFP/mNeonGreen fluorescence intensity ratios between posterior and anterior chromatids, and between QR.ap and QR.aa nuclei, are shown in blue at the lower-right corner of inverted fluorescence images. Other frames are in *Figure 3A*, and the full movies are in *Videos 3–5*. Scale bar: 2 μm. (**E**) Quantification of HDA-1::GFP, LIN-53::mNeonGreen and MYS-1::GFP fluorescence intensity ratio between QR.ap and QR.aa nuclei. Data are presented as mean ± SEM. N = 10–12. (**F**) Quantification of HDA-1::GFP (magenta), LIN-53::mNeonGreen

*Figure 1 continued on next page*

*Figure 1 continued*

(orange), and MYS-1::GFP (gray) fluorescence intensity ratios between the posterior and anterior half of QR.a or between QR.ap and QR.aa. Anaphase onset is defined as the last frame without chromatids segregation. Data are presented as mean ± SEM. N = 10–12. Statistical significance is determined by a one-sample *t*-test, with 1 as the theoretical mean. **p<0.01, ***p<0.001, ns: not significant. (G) Quantification of HDA-1::GFP (left) and LIN-53::mNeonGreen (right) fluorescence intensity ratios between the large and small daughters of cells shown on the X-axis. Data are presented as mean ± SEM. N = 7–14. Statistical significance is determined by a one-sample *t*-test, with 1 as the theoretical mean. **p<0.01, ***p<0.001.

The online version of this article includes the following figure supplement(s) for figure 1:

**Figure supplement 1.** Single-cell sequencing of the *C. elegans* embryonic cells.

**Figure supplement 2.** Asymmetric segregation of overexpressed nucleosome remodeling and deacetylase (NuRD) during asymmetric cell division (ACD) of QR.a.

**Figure supplement 3.** Asymmetric segregation of endogenous nucleosome remodeling and deacetylase (NuRD) during asymmetric cell divisions (ACDs) of Q cells.

**Figure supplement 4.** Quantifications of asymmetric nucleosome remodeling and deacetylase (NuRD) segregation during asymmetric cell division (ACD).

**Figure supplement 5.** Symmetric nucleosome remodeling and deacetylase (NuRD) segregation in *pig-1* mutant.

and surviving cells (*Conradt et al., 2016*). Intriguingly, we identified transcripts encoding subunits of the nucleosome remodeling and deacetylase (NuRD; also known as Mi-2) complex in cells where *egl-1* expression was indiscernible (*Figure 1—figure supplement 1D–F*, *Supplementary file 1*). The NuRD complex represents an evolutionarily conserved protein complex associated with chromatin that couples the activities of chromatin-remodeling ATPases with histone deacetylases (*Figure 1B*; *Bracken et al., 2019*; *Lai and Wade, 2011*). NuRD-mediated alterations in chromatin structure are vital for appropriate transcriptional regulation during cell fate determination and lineage commitment (*Bracken et al., 2019*; *Lai and Wade, 2011*), suggesting that NuRD may serve as a candidate epigenetic factor to specify cell fate between survival and death.

To investigate whether NuRD components are distributed asymmetrically between apoptotic and surviving daughter cells, we overexpressed GFP-tagged NuRD subunits within the *C. elegans* Q neuroblast lineages. Q.a neuroblast undergoes ACD to generate a large surviving Q.ap cell and a small apoptotic Q.aa cell (*Figure 1A*). We found that GFP-tagged NuRD subunits, including the histone deacetylase HDA-1, the histone binding protein LIN-53, the nucleosome remodeling factor CHD-3, and the MEP-1 protein, were enriched asymmetrically in the surviving QR.ap cell during QR.a division (*Figure 1—figure supplement 2*, *Videos 1 and 2*). Notably, NuRD asymmetric distribution occurred within several minutes after metaphase, while GFP protein translation and chromophore

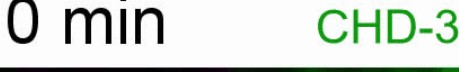

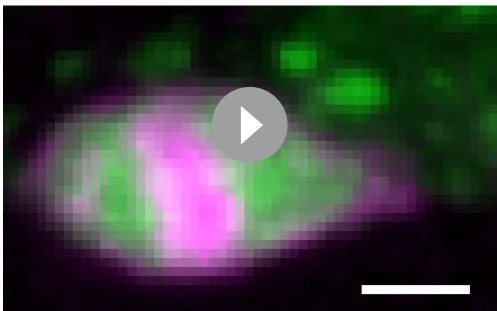

**Video 1.** Dynamics of CHD-3 during QR.a division. Fluorescence time-lapse movies of CHD-3::GFP (green) and mCherry-labeled plasma membrane and histone (magenta) in QR.a. Frames were taken every 1 min. The display rate is three frames per second. CHD-3 was asymmetrically segregated into the future surviving QR.ap. Scale bar: 2 µm.

https://elifesciences.org/articles/89032/figures#video1

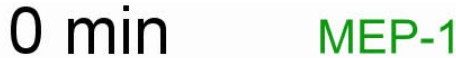

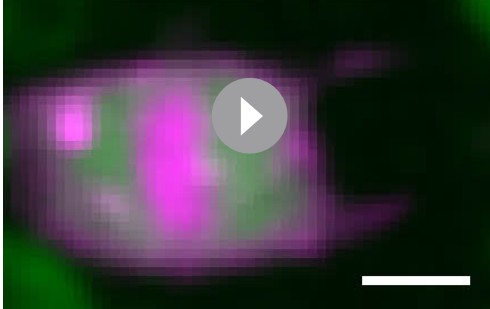

**Video 2.** Dynamics of MEP-1 during QR.a division. Fluorescence time-lapse movies of MEP-1::GFP (green) and mCherry-labeled plasma membrane and histone (magenta) in QR.a. Frames were taken every 1 min. The display rate is three frames per second. MEP-1 was asymmetrically segregated into the future surviving QR.ap. Scale bar: 2 µm.

https://elifesciences.org/articles/89032/figures#video2

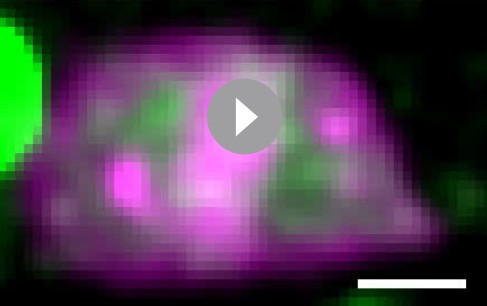

**Video 3.** Dynamics of HDA-1 during QR.a division. Fluorescence time-lapse movies of HDA-1::GFP (KI; green) and mCherry-labeled plasma membrane and histone (magenta) in QR.a. Frames were taken every 1 min. The display rate is three frames per second. HDA-1 was asymmetrically segregated into the future surviving QR.ap. Scale bar: 2 μm.

https://elifesciences.org/articles/89032/figures#video3

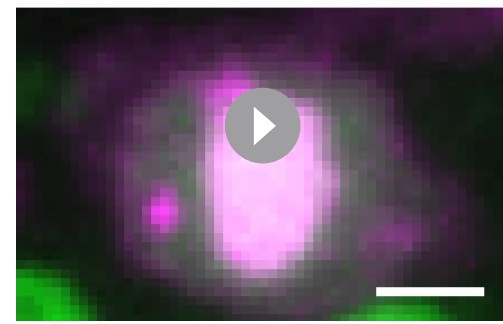

**Video 5.** Dynamics of MYS-1 during QR.a division. Fluorescence time-lapse movies of MYS-1::GFP (green) and mCherry-labeled plasma membrane and histone (magenta) in QR.a. Frames were taken every 1 min. The display rate is three frames per second. Anterior, left. MYS-1 was asymmetrically segregated into the future surviving QR.ap. Scale bar: 2 μm.

https://elifesciences.org/articles/89032/figures#video5

maturation require approximately 30 min in Q neuroblast (*Ou and Vale, 2009*), suggesting that the asymmetric enrichment of NuRD is likely the result of protein redistribution.

To investigate the dynamic distribution of endogenous NuRD during ACD, we generated a GFP knock-in (KI) strain for HDA-1 and an mNeonGreen (NG, green fluorescence) KI line for LIN-53 using CRISPR-Cas9 (*Figure 1C*). In QR.a cells, nuclear HDA-1 and LIN-53 were released into the cytoplasm and evenly distributed until metaphase (*Figure 1D*, *Videos 3 and 4*). At anaphase, HDA-1 and LIN-53 became enriched in the posterior part of QR.a but became less detectable in the anterior part (*Figure 1D*, *Figure 1—figure supplement 3A and B*, *Videos 3 and 4*). We quantified the fluorescence intensity ratio between the posterior and anterior chromatids of QR.a, and between QR.ap and QR.aa nuclei, and found that nuclear HDA-1 or LIN-53 asymmetry gradually increased from 1.1-fold at anaphase onset to 1.5- or 1.8-fold upon completion of cytokinesis, respectively (*Figure 1D and E*). We also measured the ratios of fluorescence intensities between the posterior and anterior halves of QR.a, and between QR.ap and QR.aa (*Figure 1—figure supplement 4A*, see 'Materials and methods'). NuRD asymmetry became evident at ~4 min and reached a plateau at ~6 min after the anaphase onset (*Figure 1D and F*, *Figure 1—figure supplements 3A* and *4B*). QR.a spent ~6 min from anaphase to the completion of cytokinesis (*Chai et al., 2012*; *Ou et al., 2010*), suggesting that QR.a cell establishes NuRD asymmetry during ACD.

Similar to QR.a, the dynamic and progressively developing NuRD asymmetry was also observed in the QL.a, QR.p, and QL.p cell lineages, which generate apoptotic cells (*Figure 1G*, *Figure 1—figure supplements 3B* and *4B*). In contrast, an even distribution of HDA-1 and LIN-53 was observed in two surviving daughters of QR, QL, QL.pa and QR.pa cells, which generate two viable siblings (*Figure 1G*, *Figure 1—figure supplement 3B*). To investigate whether other epigenetic factors are also asymmetrically segregated during

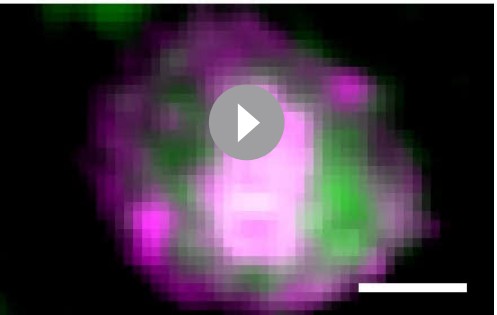

**Video 4.** Dynamics of LIN-53 during QR.a division. Fluorescence time-lapse movies of LIN-53::mNeonGreen (KI; green) and mCherry-labeled plasma membrane and histone (magenta) in QR.a. Frames were taken every 1 min. The display rate is three frames per second. LIN-53 was asymmetrically segregated into the future surviving QR.ap. Scale bar: 2 μm.

https://elifesciences.org/articles/89032/figures#video4

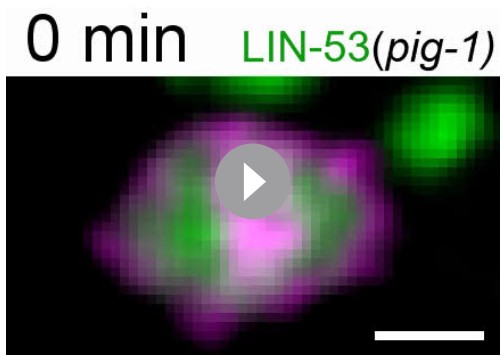

**Video 6.** Dynamics of HDA-1 during QR.a division in the *pig-1* mutant. Fluorescence time-lapse movies of HDA-1::GFP (KI; green) and mCherry-labeled plasma membrane and histone (magenta) during QR.a division in the *pig-1* mutant. Frames were taken every 1 min. The display rate is three frames per second. Anterior, left. HDA-1 was evenly segregated into two daughter cells. Scale bar: 2 µm.

https://elifesciences.org/articles/89032/figures#video6

Q cell ACDs, we tagged the MYST family histone acetyltransferase (MYS-1 in *C. elegans*) with GFP. We found that MYS-1::GFP was symmetrically segregated into the apoptotic and surviving daughter cells (*Figure 1D–F*, *Figure 1—figure supplements 3A* and *4BVideo 5*), indicating the specificity of the polarized NuRD partition. The total fluorescence of HDA-1, LIN-53, and MYS-1 remained constant during ACDs, suggesting that protein redistribution may establish NuRD asymmetry (*Figure 1—figure supplement 4C*). Inhibition of a PAR-1 family kinase PIG-1 led to symmetric Q cell divisions, producing extra neurons derived from some of their apoptotic daughters (*Cordes et al., 2006*). In *pig-1 (gm344)* mutants, HDA-1 and LIN-53 were evenly partitioned during ACDs (*Figure 1—figure supplement 5*, *Videos 6 and 7*), indicating that asymmetric NuRD segregation depends on the PIG-1 kinase.

## NuRD asymmetric segregation in embryonic cell lineages

Given that a significant portion of somatic apoptotic events in *C. elegans*-113 out of 131-occur during embryonic development (*Sulston et al., 1983*), we employed live imaging and automated cell lineage tracing algorithms (*Du et al., 2014*) to monitor the asymmetry of NuRD distribution in embryos from the early 2- or 4-cell stage up to the 350-cell stage. Our analysis indicated that in 15 out of 17 embryonic ACDs that produce apoptotic daughter cells, the surviving daughter cells showed enrichment of HDA-1 and LIN-53, with an average enrichment ratio exceeding 1.5-fold (*Figure 2A*, *Supplementary file 2*). Notably, this enrichment reached statistical significance in 6 out of 17 embryonic ACDs (*Figure 2B*). Intriguingly, the MSpppaa cell lineage's two daughter cells received comparable levels of NuRD, yet its apoptotic daughter cell completes apoptosis approximately 400 min post-division (*Hsieh et al., 2012*; *Sulston et al., 1983*), supporting the hypothesis that lower NuRD concentrations might promote apoptosis. Conversely, the apoptotic daughter cell of ABaraaaap completes apoptosis within 35 min (*Sulston et al., 1983*), despite receiving a similar amount of NuRD as its non-apoptotic sibling. Therefore, while the regulation of apoptosis in certain embryonic cells appears to be governed by NuRD-independent mechanisms, NuRD asymmetric segregation is evident in several embryonic cell lineages.

## Loss of the deacetylation activity of NuRD causes ectopic apoptosis

To investigate the role of NuRD in determining cell fate, we reduced the deacetylation activity of NuRD using RNA-mediated interference (RNAi). The *hda-1* and *lin-53* RNAi efficacy was confirmed by the reduced green fluorescence in the germlines of HDA-1::GFP and LIN-53::mNeonGreen KI animals, as well as the increased acetylation level of histone H3K27 (*Figure 3—figure supplement 1*). As reported before, RNAi of either *hda-1* or *lin-53* caused embryonic lethality (*Shi and Mello, 1998*). RNAi of *lin-53* led to embryos arrested at gastrulation stage before the generation of apoptotic cells, preventing us from analyzing its roles

**Video 7.** Dynamics of LIN-53 during QR.a division in the *pig-1* mutant. Fluorescence time-lapse movies of LIN-53::mNeonGreen (KI; green) and mCherry-labeled plasma membrane and histone (magenta) during QR.a division in the *pig-1* mutant. Frames were taken every 1 min. The display rate is three frames per second. Anterior, left. LIN-53 was evenly segregated into two daughter cells. Scale bar, 2 µm.

https://elifesciences.org/articles/89032/figures#video7

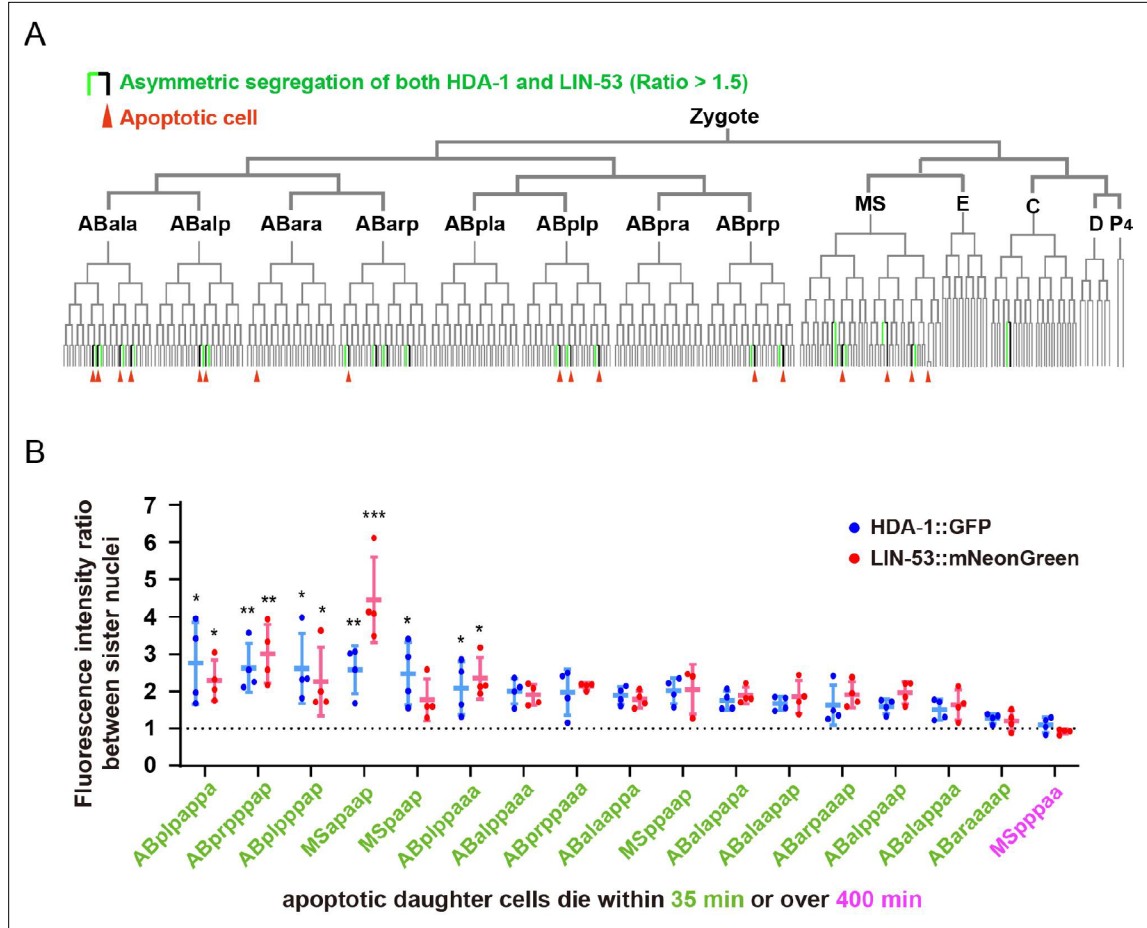

**Figure 2.** Nucleosome remodeling and deacetylase (NuRD) asymmetry in *C. elegans* embryonic cell lineages. (**A**) The tree visualization depicts the segregation of HDA-1::GFP and LIN-53::mNeonGreen between sister cells during embryonic development. In this tree structure, vertical lines represent cells and horizontal lines denote cell divisions. Green vertical lines highlight cells with higher nuclear HDA-1::GFP and LIN-53::mNeonGreen fluorescence intensity than their apoptotic sister cells (average fluorescence intensity ratio between sister cell nuclei >1.5). Red arrowheads point to apoptotic cells. The placement of cells within the tree follows the Sulston nomenclature. See also *Supplementary file 2*. (**B**) Quantifications of HDA-1::GFP and LIN-53::mNeonGreen fluorescence intensity ratios between nuclei of live daughter cells and their apoptotic sister cells. The lineage names of 17 cells that divide to produce apoptotic daughter cells are shown below the X-axis. Data are shown as mean ± SD. N = 3–4. Dunn's multiple comparisons test was used to assess statistical significance, with MSpppaa (magenta), whose apoptotic daughter cell completes apoptosis over 400 min after birth, as a control. *p<0.05, **p<0.01, ***p<0.001.

in somatic apoptosis. In contrast, *hda-1* RNAi embryos arrested between the late gastrulation stage and bean stage, allowing us to tracked some of apoptotic events. To quantify apoptotic events, we used a secreted Annexin V (sAnxV::GFP) sensor to label apoptotic cells with externalized phosphatidylserine (exPS) on the surface of the plasma membrane in *ced-1(e1735)* mutant embryos that inhibits engulfment of apoptotic cells (*Conradt et al., 2016*; *Mapes et al., 2012*; *Zhou et al., 2001*). The sAnxV::GFP labeled an average of 10 cells in *ced-1(e1735)* mutant embryos treated with control RNAi and an average of 14 cells in *ced-1(e1735); hda-1* RNAi embryos (*Figure 3A and B*). To test whether the upregulated apoptosis in *hda-1* RNAi embryos depends on the canonic apoptosis pathway, we introduced cell death-deficient mutations *ced-3/* Caspase (*n2433*), *ced-4/* APAF-1-like (*n1162*), *ced-9/* BCL-2-like (*n1950*), and *egl-1/* BH-3 only protein (*n1084n3082*) into *ced-1* mutant, respectively. RNAi of *hda-1* in these embryos only generated 0–2 exPS-positive cells (*Figure 3A and B*, *Figure 3— figure supplement 2A*), indicating that the ectopic increase of apoptosis by HDA-1 inhibition requires the EGL-1-CED-9-CED-4-CED-3 pathway. Considering the pleiotropic phenotypes caused by loss of HDA-1, we cannot exclude the possibility that ectopic cell death might result from global changes in development, even though HDA-1 may directly contribute to the life-versus-death fate determination.

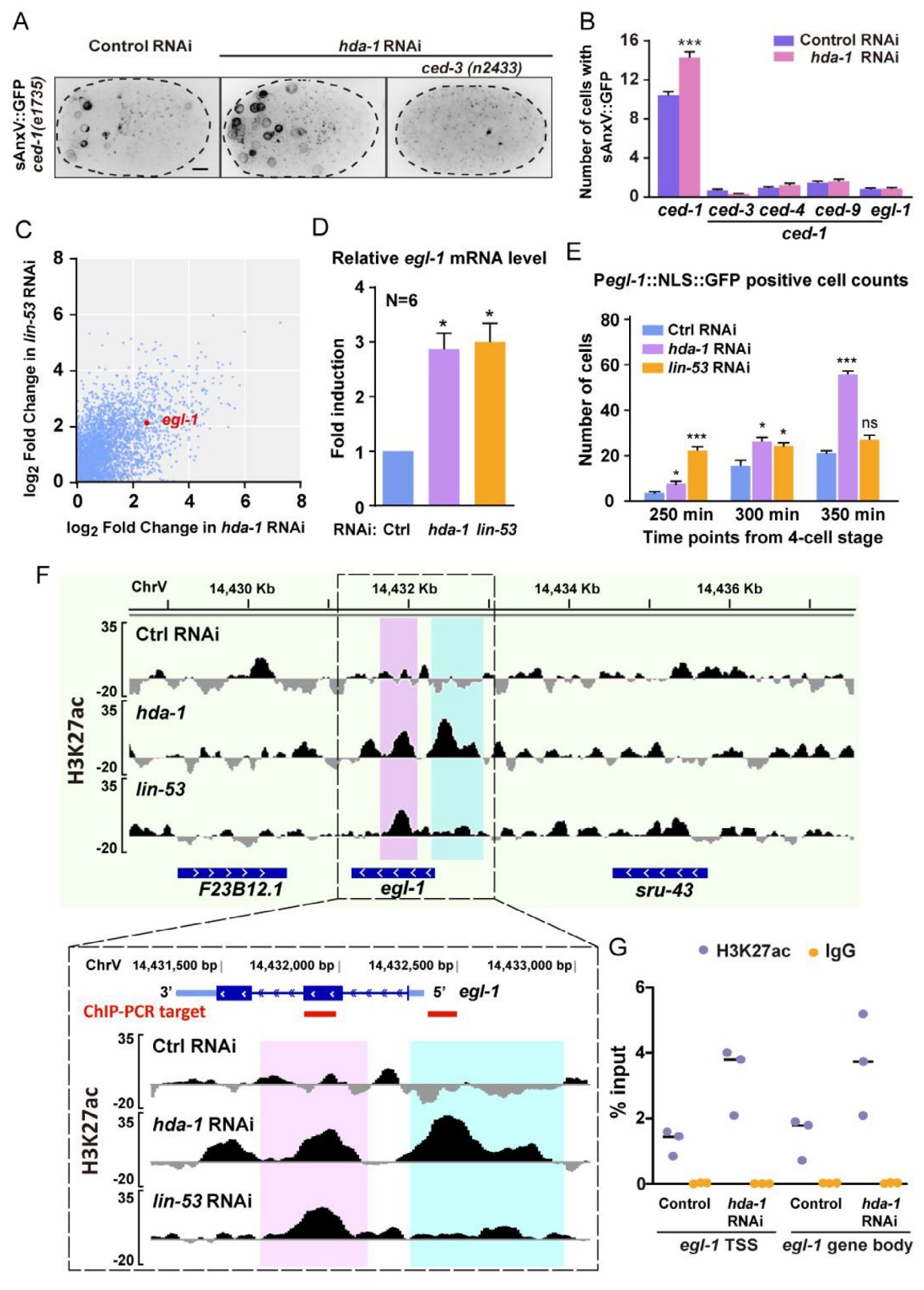

**Figure 3.** RNAi of *hda-1* induces ectopic apoptosis and increases H3K27 acetylation of the *egl-1* gene. (**A**) Representative inverted fluorescence images show P*hsp*::sAnxV::GFP from *ced-1(e1735)* and *ced-1(e1735); ced-3(n2433)* embryos between late gastrulation and bean stage, treated with control RNAi or *hda-1* RNAi. Scale bars, 5 μm. (**B**) Quantifications of cell corpse number in the embryos of indicated genotypes. Data are presented as mean ± SEM. N = 39–60. Statistical significance is determined by Student's *t*-test. ***p<0.001. (**C**) Scatter plot of the increased gene expression in both *hda-1*

*Figure 3 continued on next page*

*Figure 3 continued*

RNAi and *lin-53* RNAi animals. The *egl-1* gene is marked in red. See also **Supplementary file 3**. (**D**) Quantitative real-time PCR (RT-PCR) measurement of *egl-1* mRNA levels in the control, *hda-1*, or *lin-53* RNAi embryos. Fold induction was calculated relative to levels in control RNAi embryos. Data of six biological replicates are presented as mean ± SEM. Statistical significance is determined by the Wilcoxon test as 1 as the theoretical median. *p<0.05. (**E**) Quantification of the number of cells expressing the P*egl-1*::NLS::GFP reporter in embryos treated with control, *hda-1*, or *lin-53* RNAi. Data are presented as mean ± SEM. N = 8–9. Statistical significance is determined by Dunn's multiple comparisons test. *p<0.05, ***p<0.001, ns: not significant. (**F**) Normalized ChIP-seq signal profiles of the H3K27 acetylation level in the control, *hda-1*, or *lin-53* RNAi embryos at the *egl-1* locus. The Y-axis shows the average sequencing coverage of bins per million reads of three biological replicates normalized to the input control sample. Ectopic H3K27ac enrichments at the *egl-1* locus of *hda-1* RNAi and *lin-53* RNAi embryos are highlighted in magenta and cyan. (**G**) ChIP-qPCR analyses using H3K27ac antibody or IgG (the negative control) at selected elements of *egl-1* indicated in (**F**) by red lines. Results are shown as the percentage of input DNA. Data of three biological replicates are presented.

The online version of this article includes the following figure supplement(s) for figure 3:

**Figure supplement 1.** RNAi of *hda-1* and *lin-53* reduce fluorescence of HDA-1::GFP and LIN-53::mNeonGreen and enhance H3K27 acetylation level.

**Figure supplement 2.** HDA-1 regulates apoptotic cell fate through the canonical apoptosis pathway.

## NuRD RNAi upregulates the *egl-1* expression by increasing its H3K27 acetylation

To understand how NuRD regulates apoptotic cell fates, we performed RNA-seq analyses on WT, *hda-1* RNAi, and *lin-53* RNAi embryos. In both RNAi conditions, we observed 287 or 575 genes that were upregulated or downregulated ($|\log_2\text{FD}| > 2$ and false discovery rate [FDR] < 0.05) (**Supplementary file 3**). Notably, we found that the apoptosis-inducing gene *egl-1* was among the upregulated genes (**Figure 3C**). Our quantitative RT-PCR results further confirmed that *egl-1* transcripts were more abundant in *hda-1* RNAi and *lin-53* RNAi embryos than in control RNAi embryos (**Figure 3D**). We also used a P*egl-1*::NLS::GFP transcriptional reporter to examine the *egl-1* transcription activity during embryonic development, which allowed us to detect *egl-1* expression while avoiding the adverse effects of *egl-1* overexpression on apoptosis. We found that RNAi of *hda-1* or *lin-53* caused an increased number of P*egl-1*::NLS::GFP-positive cells (**Figure 3E**, **Figure 3—figure supplement 2B**). These results indicate that the loss of NuRD abnormally activates the *egl-1* gene expression.

Next, we investigated whether the loss of histone deacetylase activity in NuRD increased the H3K27ac levels at the *egl-1* locus as the histone H3K27ac levels are associated with transcription activation (**Heinz et al., 2015**). To test this, we performed ChIP-seq experiments with the anti-H3K27ac antibody and found that the transcription start site (TSS) and the gene body region of *egl-1* had higher levels of H3K27ac in *hda-1* or *lin-53* RNAi embryos compared to WT, while the H3K27ac levels of genes adjacent to *egl-1* showed no significant changes (**Figure 3F**). This was supported by ChIP-qPCR in *hda-1* RNAi embryos (**Figure 3G**). Therefore, inhibiting NuRD increased the H3K27ac levels at the *egl-1* locus, leading to its upregulation and subsequent activation of apoptosis.

## V-ATPase regulates asymmetric segregation of NuRD during somatic ACDs

To investigate how NuRD is asymmetrically segregated during ACD, we used affinity purification with an anti-GFP antibody and mass spectrometry (MS) to isolate NuRD binding partners from the lysate of HDA-1::GFP KI animals. Our analysis revealed all the previously known NuRD subunits from HDA-1::GFP KI animals (**Figure 4—figure supplement 1**, **Supplementary file 4**; **Bracken et al., 2019**; **Lai and Wade, 2011**), validating our experimental method. Interestingly, our co-immunoprecipitation and MS with HDA-1::GFP identified 12 subunits of the vacuolar-type H⁺-ATPase (V-ATPase) (**Figure 4A and B**), consistent with a previous report that used FLAG-tagged LIN-53 to identify 4 V-ATPase subunits (**Müthel et al., 2019**). We further confirmed the binding of HDA-1 to V-ATPase subunits in *C. elegans* by co-immunoprecipitation and western blot analysis with anti-V1A/VHA-13 antibodies (**Figure 4C**). V-ATPase is a proton pump that comprises a transmembrane domain (V0) responsible for proton transport across membranes and a peripheral ATP-hydrolytic domain (V1) that catalyzes ATP hydrolysis (**Figure 4B**). V-ATPases play a critical role in intracellular pH homeostasis and regulate numerous physiological processes, including apoptosis (**Abbas et al., 2020**; **Chen et al., 2022**; **Ernstrom et al., 2012**; **Vasanthakumar and Rubinstein, 2020**). However, the role of V-ATPase in the asymmetric segregation of epigenetic factors during ACD remains unknown.

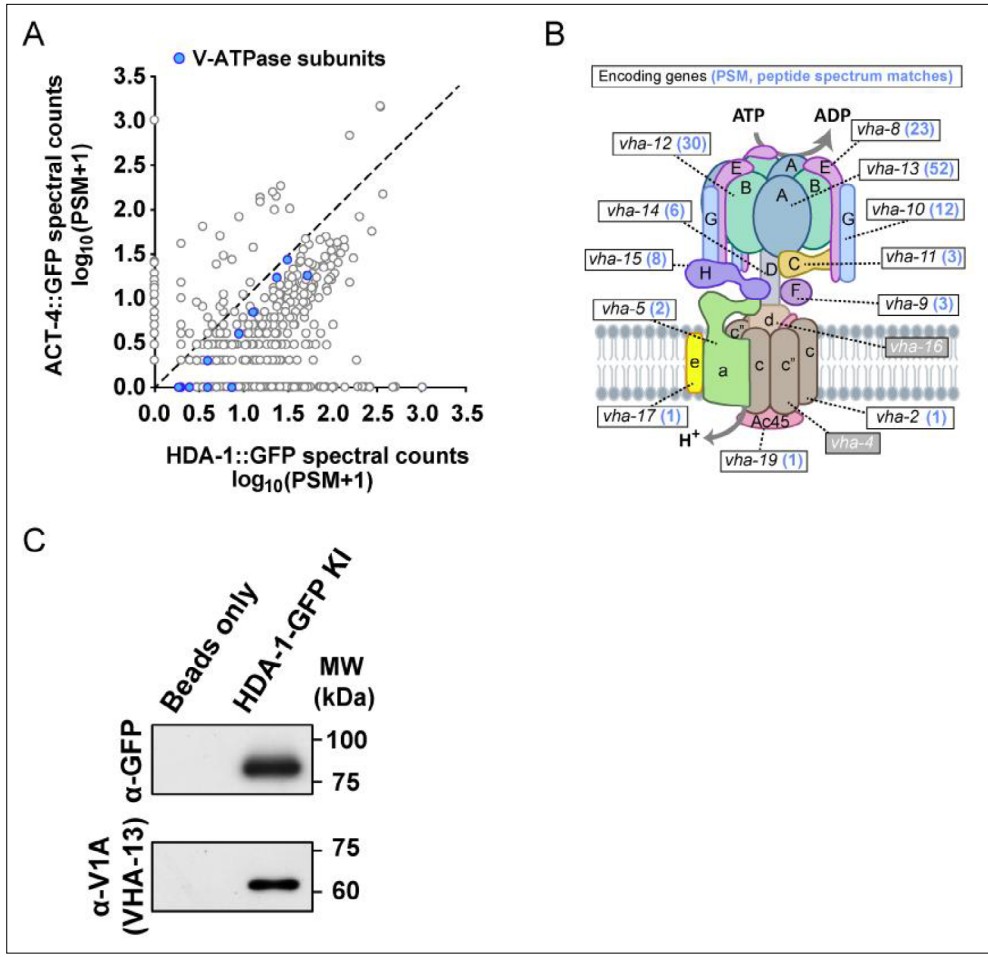

**Figure 4.** HAD-1 interacts with subunits of V-ATPase. (**A**) The plot compares counts of proteins co-precipitated with HDA-1::GFP with those with the control ACT-4 (actin)::GFP. The PSM (Peptide-Spectrum Match) is the number of identified peptide spectra matched for the protein. Blue dots represent the subunits of V-ATPase. See also *Supplementary file 4*. (**B**) Schematic model of the V-ATPase complex. The known worm subunits are indicated. The mean PSM of the encoded protein from two co-IP and MS repeats is shown in blue after the gene name. (**C**) Western blot (WB) showing co-immunoprecipitation (co-IP) of V-ATPase V1 domain A subunit (V1A) with the HDA-1 from worm lysates. Assay was performed using three biological replicates. Three independent biological replicates of the experiment were conducted with similar results.

The online version of this article includes the following figure supplement(s) for figure 4:

**Figure supplement 1.** The composition of *C. elegans* nucleosome remodeling and deacetylase (NuRD) and NuRD subunits identified by co-IP and mass spectrometry.

**Figure supplement 2.** V-ATPase co-localizes with endoplasmic reticulum.

---

To investigate the role of V-ATPase in NuRD asymmetric segregation in Q neuroblast, we used a pharmacological inhibitor, bafilomycin A1 (BafA1), to inhibit V-ATPase proton-pumping activity (*Furuchi et al., 1993*; *Wang et al., 2021*; *Yoshimori et al., 1991*). To assess the inhibitory effect of BafA1 on the proton translocation activity of V-ATPase in Q neuroblast, we monitored the cytosolic pH dynamics in dividing QR.a using the super-ecliptic pHluorin. The pHluorin is a pH-sensitive GFP reporter whose fluorescence can be quenched by the acidic pH (*Miesenböck et al., 1998*; *Sankaranarayanan et al., 2000*). Although BafA1-mediated disruption of lysosomal pH homeostasis is recognized to elicit a wide array of intracellular abnormalities, we did not observe any larval deaths and apparent abnormality in morphology at the organismal level (N > 20 for each treatment) at the dose and duration of treatment employed in this study. In DMSO-treated animals, the pHluorin fluorescence intensity remained constant in the posterior portion that forms QR.ap but was significantly reduced in the anterior portion that forms QR.aa, indicating that the cytoplasm of the future apoptotic cell

became more acidic (*Figure 5A and B*, *Video 8*). This observation is consistent with the recognition that cytosolic acidification is a common feature of both death receptor-mediated and mitochondria-dependent apoptosis (*Lagadic-Gossmann et al., 2004*; *Matsuyama et al., 2000*; *Matsuyama and Reed, 2000*). Notably, BafA1 treatment reduced the pHluorin fluorescence intensity ratio between QR.ap and QR.aa from 1.6-fold to 1.3-fold (*Figure 5A and B*, *Video 8*), suggesting that BafA1 may disrupt the cytosolic pH asymmetry in dividing QR.a cells by inhibiting V-ATPase activity, although we cannot exclude the possibility that the changes in fluorescence could be due to changes in the amount of pHluorin protein.

We observed that BafA1 treatment resulted in the symmetric segregation of HDA-1 during QR.a division (*Figure 5C and D*, *Video 9*). Notably, neither DMSO nor BafA1 treatment affected the asymmetry in daughter cell size (*Figure 5C and D*, *Video 9*), suggesting that intracellular acidification, regulated by V-ATPase activity, specifically affects the NuRD asymmetric segregation without affecting the asymmetry in daughter cell size. We also investigated whether the small QR.aa cell carrying ectopic NuRD could escape apoptosis. In DMSO-treated animals, QR.aa cells underwent apoptosis, and 12 out of 13 QR.aa cells were engulfed and degraded by a neighboring epithelial cell within 120 min after birth. However, in BafA1-treated animals, QR.aa inherited similar levels of HDA-1::GFP as its sister cell, and 11 out of 12 QR.aa cells carrying ectopic NuRD survived for over 120 min and formed a short neurite-like outgrowth (*Figure 5E*). To confirm whether the long-lived QR.aa was a consequence of the ectopic gain of NuRD, we depleted HDA-1 within Q cell lineages using the auxin-inducible protein degradation (AID) system (*Zhang et al., 2015*). By introducing the degron sequence into the *hda-1* locus and expressing the TIR1 F-box protein under the Q cell-specific *egl-17* promoter, we showed that HDA-1::GFP fluorescence in Q cells was substantially decreased under auxin treatment (*Figure 5E*). Administration of auxin restored QR.aa apoptosis in BafA1-treated animals (*Figure 5E*). These results suggest that V-ATPase activity-dependent NuRD asymmetric segregation contributes to the specification of the live-death fate.

To understand how V-ATPase regulates NuRD asymmetric segregation, we generated a transgenic strain expressing wrmScarlet-tagged VHA-17, which is the e subunit of the V0 domain. Using this strain, we were able to examine the dynamic distribution of V-ATPase during QR.a cell division. Like HDA-1, V-ATPase was also asymmetrically enriched in the surviving QR.ap portion (*Figure 6A and B*, *Video 10*). This observation suggests that V-ATPase and NuRD may co-segregate asymmetrically during ACD. We also found that BafA1 treatment disrupted V-ATPase asymmetric distribution (*Figure 6A and B*, *Video 10*), indicating the importance of the proton-pumping activity of V-ATPase in its asymmetric segregation. Therefore, our results suggest that V-ATPase may facilitate the asymmetric distribution of NuRD through its proton-pumping activity.

## Discussion

Our findings indicate that the asymmetric segregation of the NuRD complex during ACD is regulated in a V-ATPase-dependent manner and is critical for the differential expression of apoptosis activator *egl-1* and the life-versus-death fate decision (*Figure 6C*). Specifically, we have provided evidence that the reduced level of NuRD in the apoptotic daughter cell leads to an increased level of H3K27ac, an epigenetic modification known to be associated with active gene expression, at the *egl-1* locus, resulting in the upregulation of *egl-1* expression and subsequent induction of apoptosis. Based on our results, we propose a dual-function model of V-ATPase in NuRD asymmetric segregation: (i) V-ATPase interacts with NuRD, both of which are asymmetrically segregated into the surviving cell; (ii) the asymmetrical activity of V-ATPase generates a more acidic cytoplasmic environment in the future apoptotic cell portion compared to the surviving portion, which might contribute to the asymmetric segregation of V-ATPase and NuRD. This model sheds light on the mechanisms underlying the regulation of apoptosis during ACD and opens up new avenues for further investigation into the intricate interplay between V-ATPase, NuRD, and epigenetic modifications in the context of cell fate decisions.

The transcript asymmetry detected by scRNA-seq may not correspond to the protein asymmetry detected by microscopic imaging. Our scRNA-seq data shows that 6487 out of 8624 genes were not detected in *egl-1*-positive cells, the putative apoptotic cells. Cells that are *egl-1* positive may be undergoing apoptosis, rendering the asymmetry of NuRD complex transcripts insignificant in inferring protein asymmetry. Thus, the observed transcript asymmetry of the NuRD subunits between live and

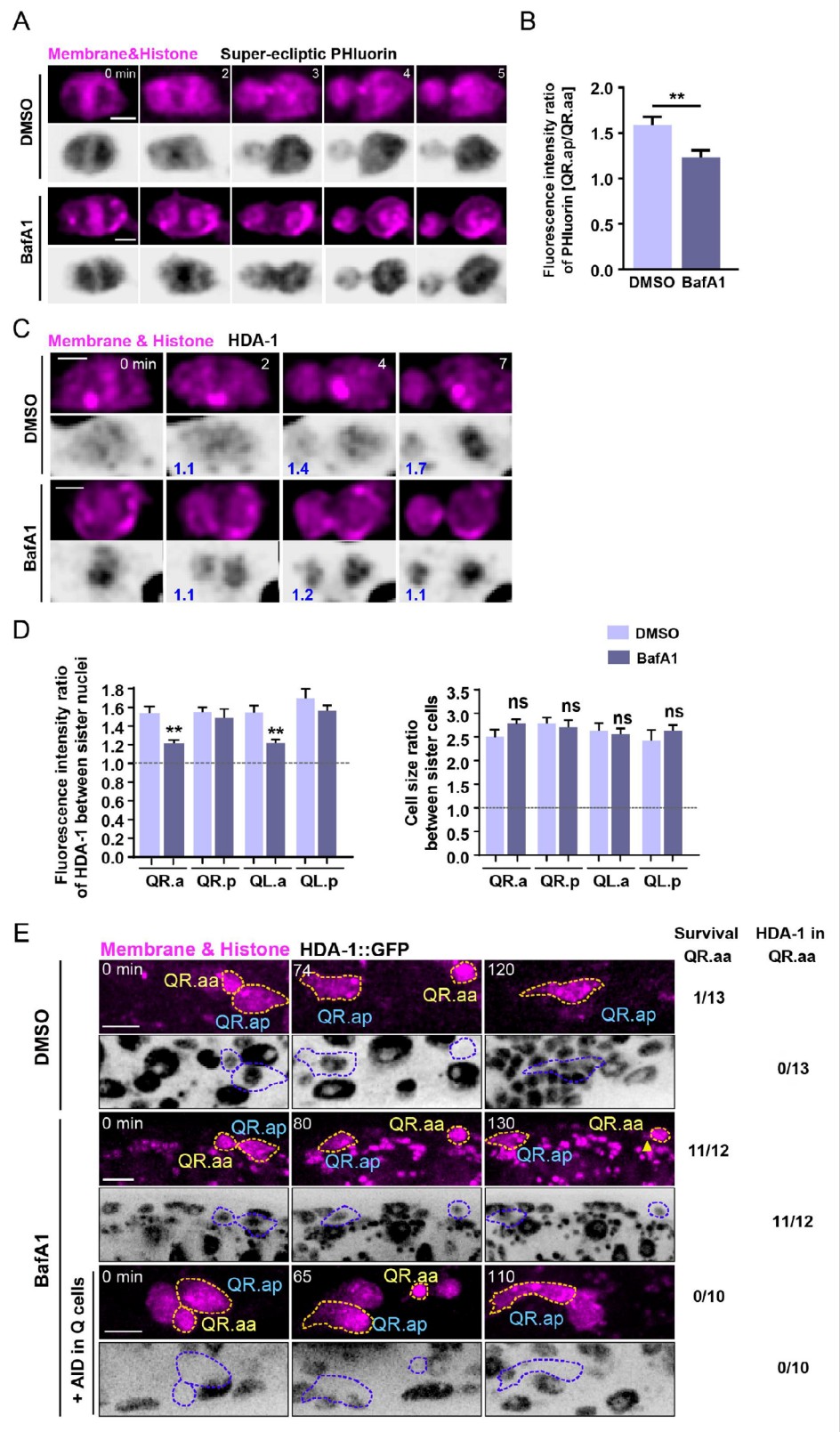

**Figure 5.** V-ATPase regulates nucleosome remodeling and deacetylase (NuRD) asymmetric segregation and cell fates. (**A**) Dynamics of the cytosolic pH indicated by super-ecliptic pHluorin during QR.a division in DMSO- or BafA1-treated animals. In each panel, the top row shows mCherry-tagged plasma membrane and histone, and the bottom row shows inverted fluorescence images of super-ecliptic pHluorin. The anterior of the cell

*Figure 5 continued on next page*

*Figure 5 continued*

is on the left. Time 0 min is the onset of anaphase. Scale bar, 2 µm. See also *Video 8*. (**B**) The super-ecliptic pHluorin fluorescence intensity ratio between QR.ap and QR.aa in DMSO control or BafA1-treated animals. Data are presented as mean ± SEM. N = 11–15. Statistical significance is determined by Student's *t*-test. **p<0.01. (**C**) Images of HDA-1::GFP distribution during QR.a cell division in DMSO- or BafA1-treated animals. In each panel, the top row shows mCherry-tagged plasma membrane and histone, and the bottom row shows inverted fluorescence images of HDA-1::GFP. The GFP fluorescence intensity ratios between the posterior and anterior chromatids, and between QR.ap and QR.aa nuclei are shown in blue at the lower left corner of inverted fluorescence images. Anterior of the cell is left. Scale bar: 2 µm. See also *Video 9*. (**D**) Quantification of HDA-1::GFP fluorescence intensity ratios between the large and small daughter cell nuclei and the cell size ratios between the large and small daughters. The names of mother cells are shown on the X-axis. Data are presented as mean ± SEM. N = 9–12. Statistical significance is determined by Student's *t*-test. **p<0.01, ns: not significant. (**E**) Representative images showing fates of QR.aa after DMSO (top), BafA1 (middle), and BafA1 plus AID treatment (bottom). The Q cell plasma membrane and chromosome are labeled by mCherry. HDA-1::GFP is shown as inverted fluorescence images. Yellow arrowhead shows a short neurite-like outgrowth of QR.aa in BafA1-treated larvae. Frequencies of QR.aa survival and HDA-1 maintenance are showed on the right. Time 0 min is the birth of QR.aa. Scale bar: 5 µm.

The online version of this article includes the following figure supplement(s) for figure 5:

**Figure supplement 1.** Quantile-quantile (Q–Q) plots for the data in *Figures 1F, G, 3B, 5B, D and 6B*.

dead cells may be coincidental with NuRD protein asymmetry during asymmetric neuroblast division, rather than serving as a regulatory mechanism.

The intrinsic mechanisms governing binary cell fate decisions involve asymmetric cortical localization of cell fate determinants, polarized partitioning of RNA species, unequal segregation of organelles, and biased distribution of damaged proteins or protein aggregates (*Sunchu and Cabernard, 2020*; *Venkei and Yamashita, 2018*). Despite this, little is known about the asymmetric segregation of epigenetic modification enzymes during ACDs, nor the functions of polarized V-ATPase distribution and cytoplasmic proton asymmetry during this process. Our findings provide new insights into the asymmetric segregation of cell-intrinsic factors and the previously unrecognized roles of V-ATPase and cytosolic acidification in this process. We cannot rule out the possibility that NuRD asymmetric segregation results from daughter cell size asymmetry. According to this perspective, the nucleus in the larger daughter cell could possess more NuRD, potentially influencing the fate of the daughter cells. However, it is important to note that the nuclear protein histone or the MYST family histone acetyltransferase is equally segregated in daughter cells of different sizes.

Two mechanisms have been proposed to explain the V-ATPase-dependent asymmetric segregation of NuRD. Firstly, a polarized intracellular transportation system could selectively deliver NuRD-V-ATPase-containing organelles to the surviving

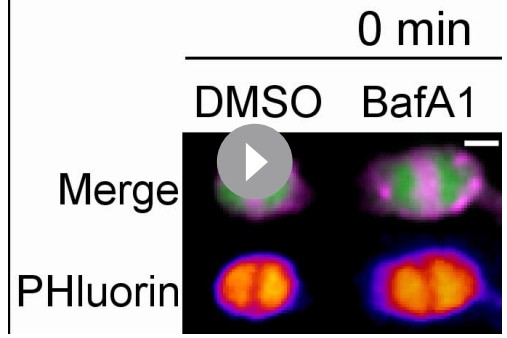

**Video 8.** Dynamics of super-ecliptic PHluorin during QR.a division. Fluorescence time-lapse movies of super-ecliptic PHluorin (green) and mCherry-labeled plasma membrane and histone (magenta) during QR.a division in DMSO and BafA1-treated animals. Frames were taken every 1 min. The display rate is three frames per second. Anterior, left. Scale bar, 2 µm.

https://elifesciences.org/articles/89032/figures#video8

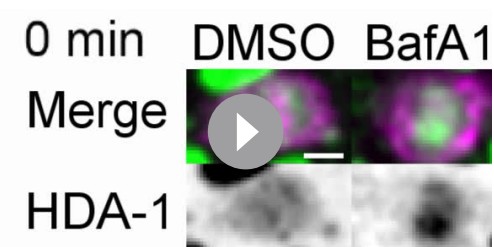

**Video 9.** BafA1 treatment disrupts HDA-1 asymmetry during QR.a division. Fluorescence time-lapse movies of HDA-1::GFP (KI; green) and mCherry-labeled plasma membrane and histone (magenta) during QR.a division in DMSO and BafA1-treated animals. Inverted fluorescence movie of HDA-1::GFP was shown below the merged movie. Frames were taken every 1 min. The display rate is three frames per second. Anterior, left. Scale bar: 2 µm.

https://elifesciences.org/articles/89032/figures#video9

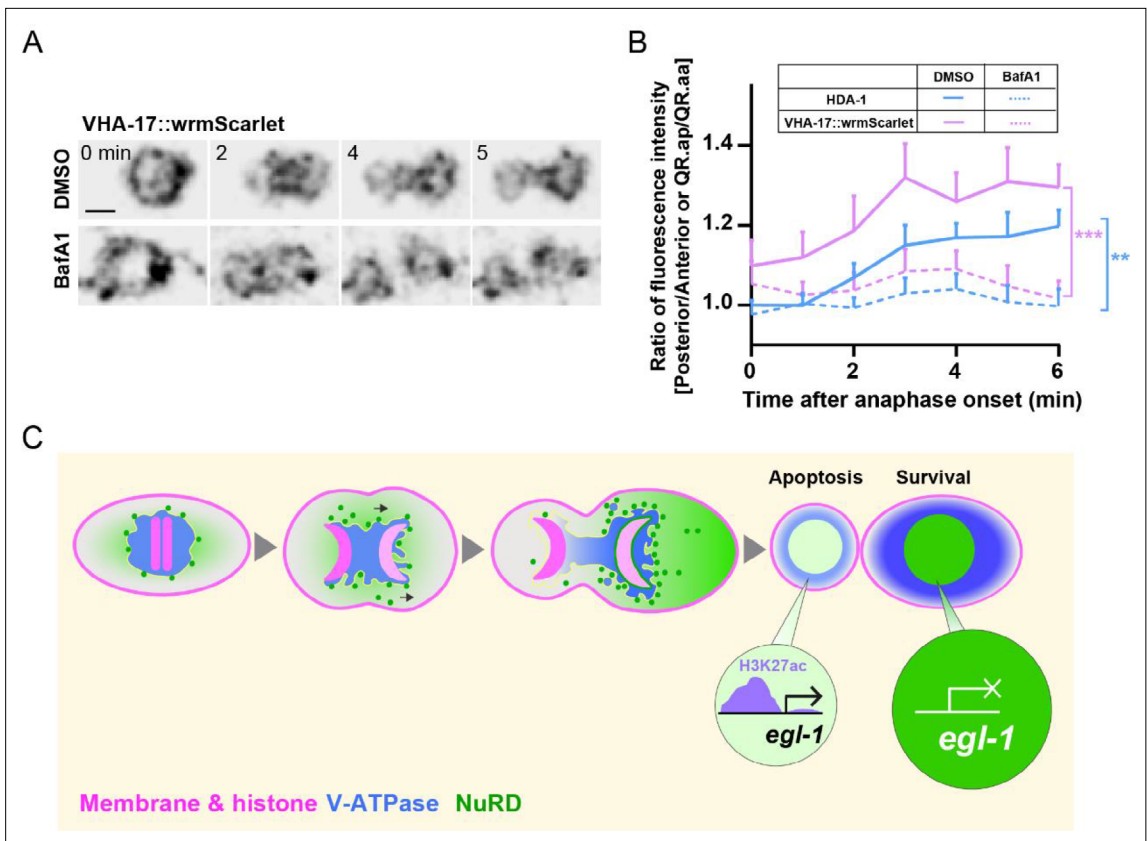

**Figure 6.** V-ATPase distribution during asymmetric cell divisions (ACDs) and a model. (**A**) Dynamics of VHA-17::wrmScarlet during QR.a cell division in DMSO- or BafA1-treated animals. VHA-17::wrmScarlet fluorescence is shown as inverted fluorescence images. Time 0 min is the onset of anaphase. Scale bar: 2 µm. See also *Video 10*. (**B**) Quantification of VHA-17::wrmScarlet (magenta) and HDA-1::GFP (blue) fluorescence intensity ratios between the posterior and anterior half of QR.a or between QR.ap and QR.aa. Data are presented as mean ± SEM. N = 10–14. Student's *t*-test was used to compare the intensity ratio difference of VHA-17 and HDA-1 between DMSO- and BafA1- treated cells at 6 min after anaphase. **p<0.01, ***p<0.001. (**C**) A proposed model. Asymmetric segregation of V-ATPase mediated the enrichment of its associated nucleosome remodeling and deacetylase (NuRD) in the large daughter cell, where high level of NuRD deacetylates *egl-1* and suppressed its expression.

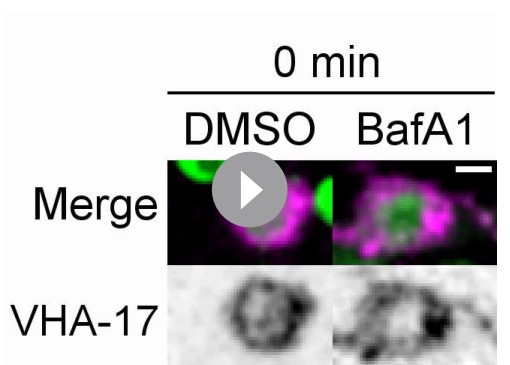

**Video 10.** Dynamics of VHA-17 during QR.a division. Fluorescence time-lapse movies of wrmScarlet-tagged VHA-17 (magenta) and HDA-1::GFP (KI; green) during QR.a division after DMSO or BafA1 treatments. Inverted fluorescence movie of VHA-17::wrmScarlet is shown below the merged movie. Frames were taken every 1 min. The display rate is three frames per second. Anterior, left. Scale bar: 2 µm.

https://elifesciences.org/articles/89032/figures#video10

daughter cell. Although V-ATPase is primarily known for its localization and function in the late endosome and lysosome, recent evidence suggests that V-ATPase subunits are synthesized and assembled on the endoplasmic reticulum (ER) (*Abbas et al., 2020*; *Graham et al., 2003*; *Wang et al., 2020*). Our findings demonstrate that V-AT-Pase colocalizes with the ER marker during ACDs, suggesting that NuRD-V-ATPase may be delivered as cargo on the ER or ER-derived vesicles (*Figure 4—figure supplement 2*). As reported in previous studies, a polarized microtubule transportation system, governed by specific motor proteins and the microtubule tracks, is responsible for the asymmetric segregation of signaling endosomes during ACDs of *Drosophila* sensory organ precursors (*Derivery et al., 2015*). It is possible that a similar biased transportation system plays a role in the asymmetric segregation of NuRD-V-ATPase-containing organelles.

Secondly, the surviving daughter cell chromatids may have a greater ability to recruit NuRD than their apoptotic sister. This could be due to a higher number of yet unknown NuRD recruiting factors on chromatids or specific post-translational modifications of chromosomal proteins that enhance the chromosomal recruitment of NuRD. Despite having an identical DNA sequence, sister chromatids can be distinguished by chromosomal proteins or modifications (*Ranjan et al., 2019*; *Tran et al., 2012*), which likely provides the specific molecular cues for asymmetric NuRD recruitment. The NuRD asymmetry on chromosomes (1.5–1.8-fold between QR.ap and QR.aa nuclei) is greater than that between the two daughter cells (approximately 1.3-fold) (*Figure 1D–F*), suggesting that polarized NuRD-V-ATPase transportation and distinct NuRD affinity from sister chromatids may act in concert to establish NuRD asymmetry.

The observation of asymmetric segregation of the NuRD complex during cell divisions suggests that the mother cell may play a role in initiating the fate specification of its daughter cells. This notion is supported by studies conducted on *Drosophila* male GSCs, which demonstrated that the old and new histone H3-H4 or unequal amounts of histone H3 variant CENP-A were incorporated into sister chromatids prior to cell division (*Ranjan et al., 2019*; *Tran et al., 2012*). Although these patterns were not observed in cell types other than GSCs, asymmetric NuRD segregation was found to be common in *C. elegans* (*Figures 1D–G* and *2A*, *Figure 1—figure supplement 3*). In 395 tracked embryonic cell divisions that generated two surviving daughter cells, NuRD was equally distributed in 390 cases, but in 5 cases, significant differences in HDA-1 and LIN-53 levels were observed between siblings (*Figure 2A*, *Supplementary file 2*). This observation suggests that biased NuRD segregation may contribute to binary cell fate decisions beyond apoptosis. In the embryonic cerebral cortex of mammals, approximately 30% of newborn cells undergo programmed cell death (*Voiculescu et al., 2000*), which is a significantly higher rate than that observed in *C. elegans* where only 12% of somatically born cells undergo apoptosis (131 apoptotic cells among 1090 somatically born cells). Thus, it is worth exploring whether brain stem cell divisions employ asymmetric NuRD segregation to specify daughter cell fates. Further studies on polarity establishment hold promise in elucidating the mechanisms underlying the asymmetric inheritance of epigenetic information.

## Materials and methods

### Worm strains and culture

*C. elegans* were maintained as described in the standard protocol (*Brenner, 1974*). All strains were cultivated at 20°C on nematode growth medium (NGM) agar plates seeded with *Escherichia coli* OP50 or HT115 (feeding RNAi assay). The wild-type strain was Bristol N2. Some strains were provided by the Caenorhabditis Genetics Center (CGC), funded by the NIH Office of Research Infrastructure Programs (P40 OD010440). The strains used in this study are listed in *Table 1*.

### Molecular biology

We performed genome editing experiments in *C. elegans* following the established methods (*Dickinson et al., 2013*; *Friedland et al., 2013*). We used the CRISPR design tool (https://zlab.bio/guide-design-resources) to select the target sites. The sgRNA sequences (*Table 2*) were inserted into the pDD162 vector (Addgene #47549) by linearizing the vector with 15 base pairs (bp) overlapped primers using Phusion high-fidelity DNA polymerase (New England Biolabs, Cat# MO531L). PCR products were digested using Dpn I (Takara, Cat# 1235A) for 2 hr at 37°C and transformed into Trans5α bacterial chemically competent cells (TransGen Biotech, Cat# CD201-01). The linearized PCR products with 15 bp overlapping ends were cyclized to generate plasmids by spontaneous recombination in bacteria. Homology recombination (HR) templates were constructed by cloning the 1.5 kb upstream and downstream homology arms into the pPD95.77 plasmid using the In-Fusion Advantage PCR cloning kit (Clontech, Cat# 639621). Fluorescence tags were inserted into the constructs with a flexible linker before the stop codons. Synonymous mutations were introduced to Cas9 target sites to avoid the cleavage of the homologous repair template by Cas9. The plasmids are listed in *Table 3*.

Constructs that express GFP-tagged NuRD components were generated using the PCR SOEing method (*Hobert, 2002*). N2 genomic sequences (1.5–2 kb promoter plus coding region) were linked with *gfp::unc-54* 3' UTR DNA fragments. Constructs that express wrmScarlet-tagged V-ATPase

**Table 1.** *C. elegans* strains in this study.

| Strain name | Genotype | Method |
|---|---|---|
| N2 | Wild-type | N.A. |
| GOU4633 | cas1133[hda-1::TEV-S::gfp knock-in] V; ujIs113[Ppie-1::H2B::mCherry, Pnhr-2::HIS-24::mCherry, unc-119(+)] II | Microinjection |
| SYS1031 | sys1031[lin-53::mNeonGreen knock-in] I; ujIs113[Ppie-1::H2B::mCherry, Pnhr-2::HIS-24::mCherry, unc-119(+)] II | Microinjection |
| GOU4279 | cas1133; casIs165[Pegl-17:: myri-mCherry, Pegl-17::mCherry-TEV-S::his-24, unc-76(+)] II | Genetic cross |
| GOU4277 | sys1031; casIs165 | Genetic cross |
| GOU4636 | casEX873[Phda-1::hda-1::gfp::unc-54 3'UTR, Pegl-17:: myri-mCherry, Pegl-17::mCherry-TEV-S::his-24] | Microinjection |
| GOU4635 | casEx874[Plin-53::lin-53::gfp::unc-54 3'UTR, Pegl-17:: myri-mCherry, Pegl-17::mCherry-TEV-S::his-24] | Microinjection |
| GOU4637 | casEx877[Pchd-3::chd-3::gfp::UNC-54 3'UTR, Pegl-17:: myri-mCherry, Pegl-17::mCherry-TEV-S::his-24] | Microinjection |
| GOU3631 | wgIs70[mep-1::TY1::EGFP::3xFLAG, unc-119(+)] III; casIs165[Pegl-17:: myri-mCherry, Pegl-17::mCherry-TEV-S::his-24, unc-76(+)] II | Genetic cross |
| GOU4634 | casEX890[Pmys-1::mys-1::gfp-unc-54UTR,Pegl-17:: myri-mCherry, Pegl-17::mCherry-TEV-S::his-24] | Microinjection |
| CU3509 | ced-1(e1735) I; smIs76[Phsp-16.41::sAnxV::gfp] | Genetic cross |
| GOU3922 | ced-3(n2433) IV; ced-1(e1735) I; smIs76[Phsp-16.41::sAnxV::gfp] | Genetic cross |
| GOU3923 | ced-4(n1162) III; ced-1(e1735) I; smIs76 | Genetic cross |
| GOU3924 | ced-9(n1950) III; ced-1(e1735) I; smIs76 | Genetic cross |
| GOU3925 | egl-1(n1084n3082) V; ced-1(e1735) I; smIs76[Phsp-16.41::sAnxV::gfp] | Genetic cross |
| smIs89 | smIs89[Pegl-1::NLS::GFP] | Microinjection |
| GOU4285 | cas1133; casIs165; pig-1(gm344) | Genetic cross |
| GOU4281 | sys1031; casIs165; pig-1(gm344) | Genetic cross |
| GOU4287 | cas1133; casEx5309[Phsp-16.2::egl-20, Pegl-17:: myri-mCherry, Pegl-17::mCherry-TEV-S::his-24] | Genetic cross |
| GOU4283 | sys1031; casEx5309[Phsp-16.2::egl-20, Pegl-17:: myri-mCherry, Pegl-17::mCherry-TEV-S::his-24] | Genetic cross |
| GOU4638 | cas1133; him-5(e1490) V | Genetic cross |
| GOU4204 | sys1031; him-5(e1490) V | Genetic cross |
| GOU4607 | cas1589 [hda-1::GSlinker::degron::TEV-S::gfp knock-in];casIs165; casEx900[Pegl-17:: TIR1::mRuby::unc-54 3'UTR,odr-1::gfp] | Microinjection |
| EX906 | casEx906 [Pegl-17::vha-17::wrmScarlet::unc-54 3'UTR;odr-1::gfp;Pegl-17::TIR1:unc-54 3'UTR];cas1589 | Microinjection |
| EX913 | casEx913[Pegl-17::sp12::gfp::unc-54 3'UTR;Pegl-17::vha-17::wrmScarlet::unc-54 3'UTR; odr-1::gfp] | Microinjection |
| SYB4702 | syb4702[gfp::rab-7 knock-in] | Microinjection: *syb4702* was generated by Suny Biotech (http://www.sunybiotech.com/) using CRISPR-Cas9 |
| EX909 | casEx909[Pegl-17::vha-17::wrmScarlet::unc-54 3'UTR;odr-1::gfp]; syb4702 | Microinjection |

*Table 1 continued on next page*

*Table 1 continued*

| Strain name | Genotype | Method |
|---|---|---|
| EX910 | casEx910 [Pegl-17::super-ecliptic PHluorin::unc-54 3'UTR; odr-1::rfp]; casIs165 | Microinjection |
| EX911 | casEx910; syb4796 | Genetic cross |

components with *gfp::unc-54* 3' UTR were generated using the In-Fusion Advantage PCR cloning kit (Clontech, Cat# 639621). The primers are listed in *Table 3*.

## Genome editing and transgenesis

To generate a knock-in strain, we purified the sgRNA construct and the repair template plasmids with the PureLink Quick PCR purification Kit (Invitrogen, #K310001) and co-injected them into N2 animals with the pRF4 [*rol-6 (su1006)*] and P*odr-1::dsRed* selection markers. The F1 transgenic progenies were singled and screened by PCR and Sanger sequencing. Transgenic *C. elegans* was generated by the germline microinjection of DNA plasmids or PCR products into N2 with P*odr-1::gfp* or P*egl-17::myri-mCherry* and P*egl-17::mCherry::his-24* plasmids. We maintained at least two independent transgenic lines with a constant transmission rate (>50%). Concentrations of DNA constructs used for generating knock-in were 50 ng/µl, and for transgenesis were 20 ng/µl. All the strains, primers, plasmids, and PCR products are listed in *Tables 1–3*, respectively.

## Live-cell imaging and quantification

Live imaging of *C. elegans* embryos, larvae, or young adults followed our established protocols (*Chai et al., 2012*; *Zhang et al., 2017*). L1 larvae or young-adult worms were anesthetized with 0.1 mmol/l levamisole in M9 buffer, mounted on 3% (wt/vol) agarose pads maintained at 20°C. Our imaging system includes an Axio Observer Z1 microscope (Carl Zeiss MicroImaging) equipped with a Zeiss 100×/1.46 numerical aperture (NA) objective, an Andor iXon+EM-CCD camera, and 488 and 561 nm lines of a Sapphire CW CDRH USB Laser System attached to a spinning disk confocal scan head (Yokogawa CSU-X1 Spinning Disk Unit). Images were acquired using µManager (https://www.micro-manager.org). Time-lapse images of Q cell divisions were acquired with an exposure time of 200 ms every 1 min. Recent advancements in optical and camera technologies permit the acquisition of Z-stacks without perturbing Q cell division or overall animal development. Z-stack images were acquired over a range of −1.6 to +1.6 µm from the focal plane, at intervals of 0.8 µm. The field of view spanned 160 µm × 160 µm, and the laser power, as measured at the optical fiber, was approximately 1 mW. ImageJ software (http://rsbweb.nih.gov/ij/) was used to perform image analysis and measurement. Image stacks were z-projected using the average projection for quantification and using the maximum projection for visual display. To follow the cell fate of QR.aa in WT, BafA1- or BafA1 and IAA-treated worms, time-lapse images were acquired with 300 ms for 561 nm and 100 ms for 488 nm exposure time at 5 min intervals. To image P*hsp-16.41*::sAnxV::GFP-positive cells in WT, mutant, or RNAi animals, we incubated the eggs at 33°C for 30 min to induce the reporter expression 3 hr

**Table 2.** Genomic targets for CRISPR.

| Gene | CRISPR-Cas9 targets (PAM) |
|---|---|
| *hda-1* knock-in | *sg1*: CTTCTACGATGGTGAGCGTGAGG |
| *hda-1* knock-in | *sg2*: GCAGCTCAGTTTGAGTCGGAAGG |
| lin-53 knock-in | *sg1*: ATTTGCGACGCGATCTTCGGAGG |
| lin-53 knock-in | *Sg2*: GGAGGTTCCATCTTCAAGAGTGG |
| vha-2 knock-in | *sg1*: ATCATTCCGACGAAGAGTCTTGG |
| vha-2 knock-in | *sg2*: GACGAAGAGTCTTGGCTGTTGGG |
| *rab-7* knock-in | *sg1*: TTTGAGCAGCGCCTTCTTTCTGG |
| *rab-7* knock-in | *sg2*: AATGTCGGGAACCAGAAAGAAGG |

**Table 3.** Plasmids and primers used in this study.

| Plasmids or PCR products | Forward primer | Reverse primer | Notes |
|---|---|---|---|
| pDD162-P*eft-3*::Cas9+P*U6*::*hda-1* sg1 | TCTACGATGGTGAGCGTGG TTTTAGAGCTAGAAATAGC | CGCTCACCATCGTAGAAG CAAGACATCTCGCAATAGGA | PCR from pDD162-P*eft-3*::Cas9+P*U6*::Empty sgRNA |
| pDD162-P*eft-3*::Cas9+P*U6*::*hda-1* sg2 | AGCTCAGTTTGAGTCGGAG TTTTAGAGCTAGAAATAGC | CGACTCAAACTGAGCTGCC AAGACATCTCGCAATAGGA | PCR from pDD162-P*eft-3*::Cas9+P*U6*::Empty sgRNA |
| pPD95.77-*hda-1*–5′ arm::3′ arm knock-in | GTACCGGTAGAAAAAAT GAACTCAAACGGCCCGTT | GGAATTCTACGAATGCGAA TAAACCCTTGCGGCTT | The 5′ arm::*hda-1*::3′ arm sequences were amplified from N2 and cloned into pPD95.77 via In-Fusion Advantage PCR Cloning Kit |
| pPD95.77-*hda-1*–5′ arm::*TEV-S*::*gfp*::3′ arm knock-in | GAACTATACAAATAGAACA CTAAAATGTGCCGCCG | CCGATCCCCCGGGCACT CTGTCTTCTGACGCTTTT | The *TEV-S-gfp* was cloned into pPD95.77-*hda-1*–5′ arm::3′ arm knock-in via In-Fusion Advantage PCR Cloning Kit |
| pPD95.77-*hda-1*::*TEV-S*::*gfp* knock-in repair template | GATGGTGAGCGTGAAGGAGAT | CTTCACGCTCACCATCGTAG | PCR on pPD95.77-*hda-1*–5′ arm::*TEV-S*::*gfp*::3′arm knock-in plasmid to synonymously mutate the PAM sequence of sg2 |
| Pegl-17:: myri-mCherry | CTTCCGTTCTATGGAACACTC | GAATCATCGTTCA CTTTTCACGG | P*egl-17* promoter was amplified from N2 genomic DNA and inserted into the pDONR P4-P1R-*mCherry* plasmid via In-Fusion Advantage PCR Cloning Kit |
| Pegl-17::mCherry ::TEV-S::his-24 | CTTCCGTTCTATGGAACACTC | GAAGACGTTGAACG TCAAATTATC | P*egl-17* promoter was amplified from N2 genomic DNA and inserted into the pDONR P4-P1R-*mCherry*::*TEV-S*::*his-24* plasmid via In-Fusion Advantage PCR Cloning Kit |
| linker::gfp::unc-54_3'UTR | AGACCCAAGCTTGGTACCA TGAGTAAAGGAGAAGAACTTTTCAC | AAGGGCCCGTACGGCC GACTAGTAGG | PCR from the plasmid pPD95.77 and then used as SOEing PCR template |
| Phda-1::hda-1 | CCAACTTCGACCTCACCCTC | GGTACCAAGCTTGGGTCTCTCT GTCTTCTGACGCTTTT | PCR from N2 genome and was then used as SOEing PCR template |
| Plin-53::lin-53 | AGCAAATGTTGCAGGGCTGTG | GGTACCAAGCTTGGGTCTCT GTTGTCTCTCTACCACAT | PCR from N2 genome and was then used as SOEing PCR template |
| Pchd-3::chd-3 | CACCTGTCCTTCGTGCCTATC | GGTACCAAGCTTGGGTCTAT ATCTCGGATAGGACGAACC | PCR from N2 genome and was then used as SOEing PCR template |
| Pmys-1::mys-1 | GCTCGTTATCAAGAAGGTCTCC | ACCAAGCTTGGGTCTGAA CATGATCTGCGCCTGAA | PCR from N2 genome and was then used as SOEing PCR template |
| Phda-1::hda-1::linker::gfp::unc-54_3'UTR | ACCAGTGCTCGACTTCGTGATG | GGAAACAGTTATGTTT GGTATATTGGG | SOEing PCR |
| Plin-53::lin-53::linker::gfp::unc-54_3'UTR | AGTCGGTCTTTGCGCTCAAC | GGAAACAGTTATGTTT GGTATATTGGG | SOEing PCR |
| Pchd-3::chd-3::linker::gfp::unc-54_3'UTR | GATCGTTGGTTAGG TCTCTCATGG | GGAAACAGTTATGTTT GGTATATTGGG | SOEing PCR |
| Pmys-1::mys-1::linker::gfp::unc-54_3'UTR | ATAAGAGCAAGAGT CAAGGCAGTC | GGAAACAGTTATGTTT GGTATATTGGG | SOEing PCR |
| egl-1 | GGCTACGAGATCGGCTCCAA | GAAGCATGGGCCGAGTAGGA | RT-qPCR |
| cdc-42 | GGAATGCTCGAGAAACTGGC | CAGTCCCTTCTGCGTCAAC | RT-qPCR |
| *egl-1* TSS region | TAATCATCCTCATCAAGCCTGC | CACAGCTTCTCATTGCACGC | ChIP-qPCR |
| *egl-1* gene body region | CTCTTCGGATCTTCTACCAATGTC | GAGTCGTCGGCAAATTGAGA | ChIP-qPCR |
| pPD95.77- Pegl-17::TIR1::mRuby::unc-54 3'UTR | ATGCAAAAGAGAATCGCCTTGT | AACAGTTATGTTTGGT ATATTGGGAATG | TIR1::mRuby::unc-54 3'UTR fragment was amplified from strain CA1210:IE28[dhc1::degron::gfp];ieSo57[Peft-3::TIR1::mRuby::unc-54 3'UTR.unc-119(+)]II |

*Table 3 continued on next page*

*Table 3 continued*

| Plasmids or PCR products | Forward primer | Reverse primer | Notes |
|---|---|---|---|
| pPD95.77- Pegl-17::vha-17::wrmScarlet::unc-54 3'UTR | cccgaaatgtgagc tATGGGTATT CTCATTCCA CTCGTC | GCTACCACTTCCAGCGTTG TTGATTACGTTTGGTGCG | *vha-17* genomic fragment was PCR from N2 genome and inserted into the pPD95.77- *Pegl-17::wrmScarlet::unc-54 3'UTR* plasmid via In-Fusion Advantage PCR Cloning Kit |
| pPD95.77-Pegl-17::sp12::gfp::unc-54 3'UTR | gcccgaaatgtgag ctATGGACG GAATGATTG CAATGC | GCTACCACTTCCAGCTTTC GTCTTCTTTGTCTCCTTTTC | *sp12* genomic fragment was PCR from N2 genome and inserted into the pPD95.77- *Pegl-17::wrmScarlet::unc-54 3'UTR* plasmid via In-Fusion Advantage PCR Cloning Kit |
| pPD95.77-Pegl-17::super-ecliptic PHluorin::unc-54 3'UTR | cccgaaatgtgagc tATGAGTAA AGGAGAAG AACTTTTCA | GAAGAGTAATTGGACCTATTTG TATAGTTCATCCATGCCA | *Super-ecliptic PHluorin* fragment was synthesized and inserted into the pPD95.77- *Pegl-17::wrmScarlet::unc-54 3'UTR* plasmid via In-Fusion Advantage PCR Cloning Kit |

post-egg-laying. After we kept the eggs at 20°C for 2 hr, we imaged them with an exposure time of 200 ms. To acquire time-lapse images of P*egl-1*::NLS::GFP-positive cells during embryonic development, we dissected 2–4-cell-stage embryos from gravid adults and imaged them with an exposure time of 300 ms every 5 min.

The quantifications of cellular fluorescence intensity ratios in Q cell lineages are described in *Figure 1—figure supplement 4A*. We used the mCherry-labeled plasma membrane to circumscribe Q cells (region of interest [ROI]). To determine the ratios of fluorescence intensities in the posterior to anterior half (P/A) of Q.a lineages or A/P of Q.p lineages, the cell in the mean intensity projection was divided into posterior and anterior halves. ImageJ software was used to measure the mean fluorescence intensities of two halves with background subtraction. The slide background's mean fluorescence intensity was measured in a region devoid of worm bodies. The background-subtracted mean fluorescence intensities of the two halves were divided to calculate the ratio. The same procedure was used to determine the fluorescence intensity ratios between two daughter cells. Total fluorescence intensity was the sum of the posterior and anterior fluorescence intensities or the sum of fluorescence intensities from two daughter cells (*Figure 1—figure supplement 4A*).

The ROIs for measuring the mean fluorescence intensities of nuclei were delineated as areas containing mCherry-tagged histone within each cell. The background-subtracted mean fluorescence intensities of the two nuclei were divided to calculate the nuclear intensity ratios.

## Feeding RNAi

RNAi bacteria were grown in LB media containing carbenicillin (50 µg/ml) and tetracycline hydrochloride (12.5 µg/ml) at 37°C overnight. RNAi bacteria were then seeded on NGM plates supplemented with 50 µg/ml carbenicillin and 1 mM isopropyl β-ᴅ-thiogalactopyranoside (IPTG). We grew the seeded RNAi plates for 2 days at room temperature, allowing double-stranded RNA expression. To score phenotypes during embryogenesis or meiosis, we transferred the synchronized young adult worms to the culture plate containing the induced *E. coli* HT115 strains expressing the control (luciferase) or specific gene-targeted dsRNAs. We incubated them for 16–24 hr at 20°C and then scored the phenotypes. As for RT-qPCR and RNA-seq experiments, parental animals of RNAi were used for laying eggs, and the progeny was collected 5 hr post-egg-laying. In ChIP-seq experiments, we used worms of mixed stages under RNAi treatment for 24–36 hr. RNAi bacterial strains were obtained from the Vidal library.

## Immunoprecipitation and mass spectrometry

GFP transgenic or knock-in worms were raised on one hundred 90-mm NGM plates. Animals were collected and washed with M9 buffer three times. For each replicate, the lysate was made from 1–2 ml packed worms with 3–4 ml of 0.5-mm-diameter glass beads using FastPrep-24 (MP Biomedicals) in lysis buffer (pH 7.4, 150 mM NaCl, 25 mM Tris-HCl, 10% glycerol, 1% NP-40, 1× cocktail of protease inhibitors from Roche [cOmplete, EDTA free], 40 mM NaF, 5 mM $Na_3VO_4$). Worm lysates were then cleared by centrifugation at 14,000 × *g* for 30 min at 4°C. For anti-GFP immunoprecipitation (IP), the

supernatant was incubated with GFP-Trap A beads (ChromoTek, GTA20) for 1 hr. Beads were then washed three times with lysis buffer. To detect the interaction between the V-ATPase subunit and HDA-1, beads are boiled with 1× SDS-PAGE sample loading buffer for SDS-PAGE and western blot. Otherwise, proteins were eluted from beads with 300 μl 0.1 M glycine-HCl (pH 2.5). Add 15 μl 1.5 M Tris-HCl (pH 8.8) to the eluted protein, followed by precipitation using 100 μl trichloroacetic acid and re-dissolution in 60 μl 8 M urea, 100 mM Tris- HCl (pH 8.5) for MS. MS samples were then reduced in 5 mM TCEP, alkylated by 10 mM iodoacetamide, and diluted fourfold with 100 mM Tris-HCl (pH 8.5). The 0.2 μg trypsin was used to digest 10 μg protein in 1 mM $CaCl_2$ and 20 mM methylamine overnight at 37°C. The resultant peptides were desalted by ZipTip pipette tips (Merck Millipore). For liquid chromatography-tandem mass spectrometry analysis, peptides were separated by an EASY-nLCII integrated nano-HPLC system (Proxeon, Odense, Denmark) directly interfaced to a Thermo Scientific Q Exactive mass spectrometer (Thermo Fisher Scientific). Peptides were loaded on an analytical fused-silica capillary column (150 mm in length, 75 mm in internal diameter; Upchurch) packed with 5 mm, 300 Å C-18 resin (Varian). Mobile phase A consisted of 0.1% formic acid, while mobile phase B consisted of 0.1% formic acid and 100% acetonitrile. The Q Exactive mass spectrometer was operated through Xcalibur 2.1.2 software in data-dependent acquisition mode. A single full-scan mass spectrum was used in the orbitrap (400–1800 m/z in mass range, 60,000 in resolution) followed by 10 data-dependent tandem mass spectrometry scans at 27% normalized collision energy (HCD). The tandem mass spectrometry spectra results were searched against the *C. elegans* proteome database by the Proteome Discoverer (version PD1.4; Thermo Fisher Scientific).

## Generation of *C. elegans* embryonic cell suspensions for SPLiT-seq

We prepared single cells from a *C. elegans* strain (*smIs89*) that expressed a P*egl-1*::NLS::GFP reporter using the established protocols with minor modifications (*Bianchi and Driscoll, 2006*; *Zhang et al., 2011*). Synchronized gravid adults were transferred to NGM plates to lay eggs for 1 hr, and about 50 μl eggs pellet were collected 3 hr post laid. We washed eggs three times with 1 ml sterile egg buffer (118 mM NaCl, 48 mM KCl, 3 mM $CaCl_2$, 3 mM $MgCl_2$, 5 mM HEPES [pH 7.2]). We rinsed the pellet with 250 μl 1 U/ml Chitinase three times. Eggshells were digested with 250 μl of 1 U/ml chitinase for 45 min on a rotter. We stopped the reaction by adding 750 μl of L-15/FBS (L-15 medium containing 10% fetal bovine serum). Eggs were pelleted by centrifugation at 900 × *g* for 3 min. The supernatant was displaced with 500 μl L-15/FBS. Embryos were dissociated by gently pipetting up and down 60 times using an insulin syringe (29G needle). The dissociated cells were separated from undissociated cell clumps by filtering through a 5 μm mesh filter, and we washed the filter with 500 μl L-15/FBS. Cells were collected by centrifuging 1 ml filtered suspension at 3220 × *g* for 3 min at 4°C. We removed the supernatant and gently resuspended cells with 2 ml 1× PBS containing 5 μl SUPERaseIn RNase Inhibitor (Invitrogen) and 2.5 μl RiboLock RNase Inhibitor (Thermo Scientific).

## SPLiT-seq library preparation and sequencing

We performed SPLiT-seq as previously described (*Rosenberg et al., 2018*) with minor modifications. After cell fixation, centrifugations were performed at 3220 × *g* for 3 min according to the *C. elegans* cell size. The RT primers were poly-dT and random hexamer mixture with Unique Molecular Identifiers (UMIs). All the barcodes used in this study were described in the published protocol (*Rosenberg et al., 2018*). The first 24 barcodes were used in the first round of barcoding, named 'Round1_01' to 'Round1_24', and 48 barcodes were used in the third round of barcoding, called 'Round3_49' to 'Round3_96'. Each cell barcode came from three rounds of barcoding. TruePrep DNA Library Prep Kit V2 constructed the sequencing library for Illumina (Vazyme Biotech #TD502) according to the manufacturer's instructions. The library was sequenced on HiSeq systems (Illumina) using 150 nucleotides (nts) kits and paired-end sequencing.

## SPLiT-seq data processing

According to the results of FastQC, adaptors or low-quality nucleotides were trimmed by Trim Galore (v0.5.2) using the default parameters. For each paired-end sequencing read, a 10 bp UMI sequence and a 24 bp cell barcode were extracted from the Read 2 file by the tool 'preprocess_splitseq.pl' of zUMIs (v0.0.6). Read 1 was split by different cell barcodes in Read 2 and mapped to the modified *C. elegans* genome (WS263 with an artificial chromosome containing the *C. elegans*

version of the GFP sequence) by zUMIs (v0.0.6) and STAR (v2.6.0c). Unique mapping reads were kept. The duplicated reads from the same transcript were excluded based on the UMI information in Read 2. Low-quality transcriptomes were removed from the analysis, yielding 442 single-cell transcriptomes, including transcript counts from 8629 genes. Among these transcriptomes, 41 transcriptomes were assigned to the putative apoptotic cell based on the expression of *egl-1* or *gfp*. Among the total 8624 genes detected in SPLiT-seq, transcripts from 6487 genes were not detected in the putative apoptotic cell population. Enriched GO terms of those 6487 genes were identified using Gene Ontology knowledgebase (*Ashburner et al., 2000*; *Carbon et al., 2021*; http://gene-ontology.org/).

## Cell lineage tracing and quantification of reporter expression

Embryo mounting, live imaging, lineage tracing, and expression quantification were performed according to a previously described procedure with minor modifications (*Bao and Murray, 2011*; *Du et al., 2014*; *Murray et al., 2008*). Two- to four-cell stage embryos were collected from young adult worms and mounted between two coverslips in the egg buffer containing 20–30 20-µm polystyrene microspheres and sealed with melted Vaseline (*Bao and Murray, 2011*). 3D time-lapse imaging was performed at 20°C ambient temperature using a spinning disk confocal microscope (Revolution XD) for 300 time points at a 75 s interval. For each time point, embryos were scanned for 30 Z focal planes with 1 µm spacing. 3D tiff stack images were processed with the StarryNite software for automated cell identification and tracing to reconstruct embryonic cell lineages using the ubiquitously expressed mCherry fluorescence. The raw results of cell identification and tracing were subjected to extensive manual inspection and editing using the AceTree program to ensure high accuracy (*Du et al., 2014*). For each mentioned strain, the cell lineage was traced from the 2- or 4-cell stage to the 350-cell stage for three or four embryos (experimental replicates). Regions of the chromatids or nuclei are determined by the H2B::mCherry signal. The fluorescent intensity of HDA-1::GFP or LIN-53::mNeonGreen in each traced nucleus at each time point was measured as the average fluorescent intensity of all the pixels within each nucleus and then with the average fluorescent intensity of the local background subtracted (*Murray et al., 2008*).

## RNA sequencing

Synchronized young adult worms were cultured on RNAi plates for 24 hr. Gravid adults were transferred to new RNAi plates to lay eggs for 1 hr. Eggs were harvested 5 hr post-laid and then lysed with TRIzol reagent (Invitrogen). We extracted the total RNA following the manufacturer's protocol. The Qubit RNA High Sensitivity Assay Kit (Invitrogen) was used to quantify RNA concentration. The Agilent 2100 bioanalyzer system was used for the assessment of RNA quality. We used samples with an RNA integrity number (RIN) above 6.0 for sequencing library construction. We used identical input total RNA (50–500 ng) between control and samples for library preparation using the KAPA RNA HyperPrep Kit (KAPA Biosystems, Wilmington, MA). Libraries were analyzed by Agilent 2100 bioanalyzer system for quality control. The library samples were sequenced on an Illumina NovaSeq 6000 platform. Approximately 5 GB of raw data were generated from each sample with 150 bp paired-end read lengths.

## RNA-seq data analysis

FastQC assessed the quality score, adaptor content, and duplication rates of sequencing reads. Trim_galore (v0.6.0) was used to remove the low-quality bases and adaptor sequences with default parameters. After trimming, paired-end reads with at least 20 nucleotides in length were aligned to the *C. elegans* reference genome (WS263) using STAR (2.5.4b) with default parameters. Read counts representing gene expression levels were calculated with HTSeq (version 0.9.1) using default parameters. Uniquely mapped reads were included. Gene names were annotated on the Wormbase website (https://wormbase.org/tools/mine/simplemine.cgi). DESeq2 package in R programming language was used for differential expression analysis. Differentially expressed genes were defined as upregulated genes (log2-transformed fold change greater than 1) or downregulated genes (log2-transformed fold change less than –1) with a FDR <0.05. We used the ggplot2 package of R to plot figures.

## Reverse transcription and quantitative real-time PCR

Young adult worms were cultured on RNAi plates for 24 hr. We transferred the gravid adults to new RNAi plates to lay eggs for 1 hr. Eggs were harvested 5 hr post-laid and then lysed with TRIzol reagent (Invitrogen). The RNeasy Mini kit (QIAGEN) prepared the total RNA from ~20 µl egg pellets for each biological replicate. cDNA from the same amount of RNA between RNAi and control samples were synthesized in a 20 µl reaction volume by the PrimeScript RT reagent Kit with gDNA Eraser (TAKARA, Code No. RR047A). 1 µl of cDNA was used as the template in a 10 µl reaction volume of the PowerUp SYBR Green Master Mix (Applied Biosystems) with four technical replicates. Quantitative real-time PCR was performed in two independent experiments with three biological replicates each time using the Applied Biosystems QuantStudio 1 Real-Time PCR System and normalized to the cell division cycle-related GTPase encoding gene *cdc-42*. Data were analyzed using the standard curve method. The primers used in real-time PCR assays are listed in *Table 3*.

## Chromatin immunoprecipitation

For ChIP assays, chromatin immunoprecipitation was performed as previously described (*Mukhopadhyay et al., 2008*) with modifications. Mixture stage worms on RNAi plates were harvested and washed three times with M9 buffer. 0.4 ml of packed worms were obtained and frozen into small balls with liquid nitrogen for each ChIP replicate. We quickly ground the little worm balls into powders in liquid nitrogen. Worm powder was crosslinked in 4 ml crosslinking buffer (1.1% formaldehyde in PBS with protease and phosphatase inhibitors) with constant rotation for 15 min at room temperature. Fixed worm samples were quenched by incubation with 0.125 M glycine for 5 min and then washed in cold PBS-PIC (PBS buffer with protease inhibitor cocktail) three times. The tissue pellet resuspended in PBS-PIC was homogenized by 2–3 strokes in a Dounce homogenizer. The cell suspension was centrifuged at 2000 × *g* for 5 min at 4°C, and we removed the supernatant.

We performed nuclei preparation, chromatin digestion, chromatin immunoprecipitation, and DNA Purification using SimpleChIP Enzymatic Chromatin IP Kit (Magnetic Beads) (Cell Signaling Technology, #9003). Samples were sonicated using a Qsonica sonicator (Qsonica Q800R) at 70% amplitude for five cycles of 30 s on and 30 s off with four repeats. The chromatin was digested using 6 µl of micrococcal nuclease (MNase) (CST Cat# 10011S) in 300 µl of buffer B (CST Cat# 7007) containing 0.5 mM DTT for 8 min at 37°C. The quality of chromatin digestion was analyzed by Agilent 2100 bioanalyzer system. Before IP, the chromatin sample concentration was assessed using the BCA Protein Assay Kit (Tiangen, Cat# PA115-02). 1 µg H3K27ac antibody (Abcam ab4729) or 2 µg IgG was used per IP with 0.5–1 mg input chromatin protein.

According to the manufacturer, we used 200 ng purified DNA samples to construct the ChIP-seq library using the VAHTS Universal DNA Library Prep Kit for Illumina V3 (Vazyme ND607). DNA libraries were sequenced on an Illumina NovaSeq 6000 platform, with 150 bp paired-end sequencing. Three biological replicates were collected for two independent ChIP-seq experiments. For ChIP-qPCR analysis of H3K27ac enrichment in the *egl-1* locus, the same volume of cDNA from the IP experiment described above was used for qPCR analysis. *Table 3* lists the primers.

## ChIP-seq data processing

150 bp double-end reads were aligned to the *C. elegans* genome version WBcel235 using Bwa-mem2 version 2.1 with default parameters. Only uniquely mapped reads were kept. Samples were normalized by bins per million mapped reads (BPM) and averaged, and then treated samples were subtracted by input samples to calculate enrichment using deepTools2 (*Ramírez et al., 2016*). Data were visualized using IGV (*Robinson et al., 2011*).

## Immunofluorescence

We performed immunofluorescence of *C. elegans* eggs using the freeze-cracking method. In brief, eggs were washed with M9 buffer three times and dripped onto poly-L-lysine-coated slides. After freezing in liquid nitrogen for 5 min, the coverslip was swiftly removed, and embryos were fixed in methanol at –20°C for 15 min. We washed the slides twice with PBST (PBS + 0.05% Tween 20). We incubated slides with 1% BSA in PBST for 30 min at room temperature to block the unspecific bindings. The primary and secondary antibodies were diluted in PBST plus 1% BSA by 1:500-1:2000 for primary antibodies (ab10799 for histone H3 and ab4729 for H3K27ac) and 1:5000 for the secondary

antibodies (Alexa-488 anti-mouse and Alexa546 anti-rabbit from Invitrogen). The primary antibody was incubated overnight at 4°C in a moisture chamber, and the secondary antibody was incubated for 1 hr at room temperature. After washing with PBST, we applied 1–2 drops of the Fluoroshield Mounting Medium containing DAPI (Abcam, ab104139) and coverslips to the samples.

## Pharmacological and chemical treatments

To block V-ATPase proton pump activity in the L1-L2 larvae, which have cuticle layers that may compromise agent penetration, the Bafilomycin A1 treatment was designed based on the published protocols (*Kumsta et al., 2017*; *Papandreou and Tavernarakis, 2017*; *Pivtoraiko et al., 2010*). Briefly, gravid worms were placed in one drop of ddH$_2$O containing 3% (v/v) pelleted OP50 and 100 µM bafilomycin A1 (Abcam, dissolved in DMSO) for 16 hr at 20°C. After incubation, L1 larvae were washed with M9 buffer and recovered for 5 h on NG agar plates before imaging.

To achieve Q cell conditional HDA-1 depletion, the auxin-inducible degradation (AID) was performed as described previously (*Zhang et al., 2015*). The natural auxin (indole-3-acetic acid [IAA]) was purchased from Alfa Aesar (#A10556). A 400 mM solution in ethanol was stored at 4°C as stock. Before imaging, L1 larvae were transferred to the S basal buffer supplemented with 3% (v/v) pelleted OP50 and 4 mM of auxin for a 1.5 hr treatment.

## Coimmunoprecipitation and western blot

For immunoprecipitation of endogenous V-ATPase A subunit with HDA-1, HDA-1-GFP knock-in worms were lysed in ice-cold lysis buffer (pH 7.4, 150 mM NaCl, 25 mM Tris-HCl, 10% glycerol, 1% NP-40, 2× cocktail of protease inhibitors from Roche [cOmplete, EDTA free], 40 mM NaF, 5 mM Na$_3$VO$_4$). The soluble fractions from worm lysates were immunoprecipitated with anti-GFP GFP-Trap A beads (ChromoTek, GTA20) for 1 hr at 4°C. Immunoprecipitates were washed three times with lysis buffer containing 1× cocktail of protease inhibitors from Roche (cOmplete, EDTA free) and were then transferred to new tubes. Beads were washed again with 100 mM PB (pH 6.0, 6.5, 7.0, or 7.5) and boiled in 1× SDS loading buffer under 95°C for 5 min. The protein samples were analyzed by western blotting.

Protein samples were resolved by SDS-PAGE and transferred to PVDF membranes. Each membrane was divided into two parts to be incubated with GFP (Abcam ab290, 1:4000) or ATP6V1A (Abcam ab199326, 1:2000) primary antibodies, followed by HRP-conjugated Goat Anti-Rabbit IgG (H&L) (Easybio, 1:5000) secondary antibodies. Quantification of the endogenous V-ATPase A subunit interacting with HDA-1 was performed using HDA-1::GFP bands as a calibration standard. Quantitative densitometry of chemiluminescent bands was performed using ImageJ software.

## Acknowledgements

We thank Drs. W Zhong, G Garriga, D Xue, G Li, and Y Li and for discussion. This study was supported by the National Key R&D Program of China to WL and GO (2022YFA1302700, 2017YFA0102900, 2019YFA0508401, 2017YFA0503501), and the National Natural Science Foundation of China to WL, GO, and YC (grants 32270773, 32070706, 31991190, 31730052, 31525015, 31861143042, 31561130153, 31671451).

## Additional information

### Funding

| Funder | Grant reference number | Author |
|---|---|---|
| National Key Research and Development Program of China | 2022YFA1302700 | Wei Li |
| National Key Research and Development Program of China | 2017YFA0503501 | Guangshuo Ou |

| Funder | Grant reference number | Author |
|---|---|---|
| National Key Research and Development Program of China | 2017YFA0102900 | Wei Li |
| National Key Research and Development Program of China | 2019YFA0508401 | Guangshuo Ou |
| National Natural Science Foundation of China | 32270773 | Wei Li |
| National Natural Science Foundation of China | 32070706 | Wei Li |
| National Natural Science Foundation of China | 31991190 | Guangshuo Ou |
| National Natural Science Foundation of China | 31730052 | Guangshuo Ou |
| National Natural Science Foundation of China | 31525015 | Guangshuo Ou |
| National Natural Science Foundation of China | 31861143042 | Guangshuo Ou |
| National Natural Science Foundation of China | 31671451 | Yongping Chai |
| National Natural Science Foundation of China | 31561130153 | Guangshuo Ou |

The funders had no role in study design, data collection and interpretation, or the decision to submit the work for publication.

## Author contributions

Zhongyun Xie, Resources, Data curation, Formal analysis, Investigation, Writing - original draft, Project administration, Writing – review and editing; Yongping Chai, Data curation, Funding acquisition; Zhiwen Zhu, Resources, Methodology; Zijie Shen, Resources, Investigation, Methodology; Zhengyang Guo, Validation; Zhiguang Zhao, Long Xiao, Zhuo Du, Resources, Investigation; Guangshuo Ou, Conceptualization, Supervision, Funding acquisition, Writing - original draft; Wei Li, Supervision, Validation, Project administration, Writing – review and editing

## Author ORCIDs

Zhongyun Xie http://orcid.org/0000-0003-3906-1217
Zhengyang Guo http://orcid.org/0009-0001-2746-613X
Zhuo Du http://orcid.org/0000-0002-6322-4656
Wei Li http://orcid.org/0000-0001-9133-461X

Joint public review: https://doi.org/10.7554/eLife.89032.4.sa1
Author response https://doi.org/10.7554/eLife.89032.4.sa2

# Additional files

## Supplementary files

• Supplementary file 1. Single-cell SPLiT-seq expression matrix and genes that were not detected from the P*egl-1-NLS-gfp*-positive cells.

• Supplementary file 2. The fluorescence intensity ratio of each cell pair and normalized fluorescence intensity of each cell in embryonic lineage tracing assay.

• Supplementary file 3. Differential expression of RNA-seq under control, *hda-1*, or *lin-53* RNAi.

• Supplementary file 4. Anti-GFP IP-MS results of HDA-1::GFP KI worms and ACT-4::GFP transgenic worms.

• MDAR checklist

## Data availability

The data sets generated and analyzed in this study are available in the NCBI Gene Expression Omnibus (GEO, http://www.ncbi.nlm.nih.gov/geo) under accession number GSE167379.

The following dataset was generated:

| Author(s) | Year | Dataset title | Dataset URL | Database and Identifier |
|---|---|---|---|---|
| Xie Z, Chai Y, Li M, Ou G | 2022 | Asymmetric Segregation of the Nucleosome Remodeling and Deacetylase Complex during Asymmetric Cell Divisions | https://www.ncbi.nlm.nih.gov/geo/query/acc.cgi?acc=GSE167379 | NCBI Gene Expression Omnibus, GSE167379 |

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
