## [Editor Report · eLife assessment]

The authors make the intriguing proposal that the NuRD complex in *C. elegans*, which has been linked to the regulation of the cell death protein EGL-1 before, becomes asymmetrically distributed after cell division and that this asymmetry relies on V-ATPase activity. Whereas some disagreement remained between the reviewers' and the authors' interpretation, the final version incorporated alternative possibilities in the text, and with careful interpretation, the current article's model is supported by **solid** data, and represents a **valuable** contribution to the field.

---

## [Referee Report · Joint public review]

Xie et al. propose that the asymmetric segregation of the NuRD complex is regulated in a V-ATPase-dependent manner, and plays a crucial role in determining the differential expression of the apoptosis activator egl-1 and thus critical for the life/death fate decision.

Remaining concerns are the following:

The authors should provide the point-by-point response to the following issues. In particular, authors should provide clear reasoning as to why they did not address some of the following comments in the previous revisions. The next response should be directly answering to the following concerns.

(1) Discussion should be added regarding the criticism that NuRD asymmetric segregation is simply a result of daughter cell size asymmetry. It is perfectly fine that the NuRD asymmetry is due to the daughter cell size difference (still the nucleus within the bigger daughter would have more NuRD, which can determine the fate of daughter cells). Once the authors add this clarification, some criticisms about 'control' may become irrelevant.

(2) ZEN-4 is a kinesin that predominantly associates with the midzone microtubules and a midbody during mitosis. Given that midbodies can be asymmetrically inherited during cell division, ZEN-4 is not a good control for monitoring the inheritance of cytoplasmic proteins during asymmetric cell division. Other control proteins, such as a transcriptional factor that predominantly localizes in the cytoplasm during mitosis and enters into nucleus during interphase, are needed to clarify the concern.

As for pHluorin experiments, symmetric inheritance of GFP and mCherry is not an appropriate evidence to estimate the level of pHluorin during asymmmetric Q cell division. This issue remains unsolved.

(3) Q-Q plot (quantile-quantile plot) in Figure S10 can be used for visually checking normality of the data, but it does not guarantee that the distribution of each sample is normal and has the standard deviation compared with the other samples. I recommend the authors to show the actual statistical comparison P-values for each case. The authors also need to show the number of replicate experiments for each figure panel.

The authors left inappropriate graphs in the revised manuscript. In Figure 3E, some error bars are disconnected and the other are stuck in the bars. In Figure S4C, LIN-53 in QR.a/p graph shows lines disconnected from error bars.

I am bit confused with the error bars in Figure 2B. Each dot represents a fluorescent intensity ratio of either HDA-1 or LIN-53 between the two daughter cells in a single animal. Plots are shown with mean and SEM, but several samples (for example, the left end) exhibit the SEM error bar very close to a range of min and max. I might misunderstand this graph but am concerned that Figure 2B may contain some errors in representing these data sets. I would like to ask the authors to provide all values in a table format so that the reviewers could verify the statistical tests and graph representation.

(4) The authors still do not provide evidence that the increase in sAnxV::GFP and Pegl-1gfp or the increase in H3K27ac at the egl-1 gene in hda-1(RNAi) and lin-53(RNAi) animals is not a consequence of global effects on development. Indeed, the images provided in Figure S7B demonstrate that there are global effects in these animals. no causal interactions have been demonstrated.

(5) Figure 4: Due to the lack of appropriate controls for the co-IP experiment (Fig. 4), I remain unconvinced of the claim that the NuRD complex and V-ATPase specifically interact. Concerning the co-IP, the authors now mention that the co-IP was performed three times: "Assay was performed using three biological replicates. Three independent biological replicates of the experiment were conducted with similar results." However, the authors did not use ACT-4::GFP or GFP alone as controls for their co-IP as previously suggested. This is critical considering that the evidence for a specific HDA-1::GFP - V-ATPase interaction is rather weak (compare interactions between HDA-1::GFP and V-ATPase subunits in Fig 4B with those of HDA-1::GFP and subunits of NuRD in Fig S8B).

(6) Based on Fig 5E, it appears that Bafilomycin treatment causes pleiotropic effects on animals (see differences in HDA-1::GFP signal in the three rows). The authors now state: "Although BafA1-mediated disruption of lysosomal pH homeostasis is recognized to elicit a wide array of intracellular abnormalities, we found no evidence of such pleiotropic effects at the organismal level with the dosage and duration of treatment employed in this study". However, the 'evidence' mentioned is not shown. It is critical that the authors provide this evidence.

---

## [Author Response]

The following is the authors’ response to the current reviews.

**Joint Public Review:**
Xie et al. propose that the asymmetric segregation of the NuRD complex is regulated in a V-ATPase-dependent manner, and plays a crucial role in determining the differential expression of the apoptosis activator egl-1 and thus critical for the life/death fate decision.Remaining concerns are the following:The authors should provide the point-by-point response to the following issues. In particular, authors should provide clear reasoning as to why they did not address some of the following comments in the previous revisions. The next response should be directly answering to the following concerns.(1) Discussion should be added regarding the criticism that NuRD asymmetric segregation is simply a result of daughter cell size asymmetry. It is perfectly fine that the NuRD asymmetry is due to the daughter cell size difference (still the nucleus within the bigger daughter would have more NuRD, which can determine the fate of daughter cells). Once the authors add this clarification, some criticisms about 'control' may become irrelevant.

We thank the reviewer for this suggestion. We will add the following text in the revised discussion on page 14, line 26:

“…We cannot rule out the possibility that NuRD asymmetric segregation results from daughter cell size asymmetry. According to this perspective, the nucleus in the larger daughter cell could possess more NuRD, potentially influencing the fate of the daughter cells. However, it is important to note that the nuclear protein histone or the MYST family histone acetyltransferase is equally segregated in daughter cells of different sizes.….”

(2) ZEN-4 is a kinesin that predominantly associates with the midzone microtubules and a midbody during mitosis. Given that midbodies can be asymmetrically inherited during cell division, ZEN-4 is not a good control for monitoring the inheritance of cytoplasmic proteins during asymmetric cell division. Other control proteins, such as a transcriptional factor that predominantly localizes in the cytoplasm during mitosis and enters into nucleus during interphase, are needed to clarify the concern.

We clarified the issue of ZEN-4 below:

The critique assumes that "midbodies can be asymmetrically inherited during cell division." However, this assumption does not apply to our study of Q cell asymmetric divisions. In our earlier research, we demonstrated that midbodies in Q cells are released post-division and subsequently engulfed by surrounding epithelial cells (Chai et al., Journal of Cell Biology, 2012). Moreover, we have shown that midbodies from the first cell division in *C. elegans* embryos are also released and engulfed by the P1 cell (Ou et al., Cell Research, 2013). Therefore, the notion of midbody asymmetric inheritance is irrelevant to this manuscript. Additionally, our manuscript already presents the example of the MYST family histone acetyltransferase, illustrating a nuclear protein that predominantly localizes in the cytoplasm during mitosis and symmetrically enters the nucleus during interphase.

As for pHluorin experiments, symmetric inheritance of GFP and mCherry is not an appropriate evidence to estimate the level of pHluorin during asymmmetric Q cell division. This issue remains unsolved.

We acknowledge the limitation of pHluorin in measuring the pH level in a living cell. Future studies could be performed to measure the dynamics of pH levels when advanced tools are available.

(3) Q-Q plot (quantile-quantile plot) in Figure S10 can be used for visually checking normality of the data, but it does not guarantee that the distribution of each sample is normal and has the standard deviation compared with the other samples. I recommend the authors to show the actual statistical comparison P-values for each case. The authors also need to show the number of replicate experiments for each figure panel.

We thank the reviewer for pointing this out. We will provide P-values for each case and the number of replicate experiments in the revised Figure 5-figure supplement 1 ( corresponding to Figure S10) and the figure legend.

The authors left inappropriate graphs in the revised manuscript. In Figure 3E, some error bars are disconnected and the other are stuck in the bars. In Figure S4C, LIN-53 in QR.a/p graph shows lines disconnected from error bars.

We thank the reviewer for pointing this out. We will correct these error bars.

I am bit confused with the error bars in Figure 2B. Each dot represents a fluorescent intensity ratio of either HDA-1 or LIN-53 between the two daughter cells in a single animal. Plots are shown with mean and SEM, but several samples (for example, the left end) exhibit the SEM error bar very close to a range of min and max. I might misunderstand this graph but am concerned that Figure 2B may contain some errors in representing these data sets. I would like to ask the authors to provide all values in a table format so that the reviewers could verify the statistical tests and graph representation.

We thank the reviewer for pointing this out. We apologize for the typo in Figure 2B figure legend. We will correct SEM to SD.

(4) The authors still do not provide evidence that the increase in sAnxV::GFP and Pegl-1gfp or the increase in H3K27ac at the egl-1 gene in hda-1(RNAi) and lin-53(RNAi) animals is not a consequence of global effects on development. Indeed, the images provided in Figure S7B demonstrate that there are global effects in these animals. no causal interactions have been demonstrated.

We cannot exclude the global effects and have discussed this issue in our previous manuscript on page 9, line 26:

“...Considering the pleiotropic phenotypes caused by loss of HDA-1, we cannot exclude the possibility that ectopic cell death might result from global changes in development, even though HDA-1 may directly contribute to the life-versus-death fate determination.”

(5) Figure 4: Due to the lack of appropriate controls for the co-IP experiment (Fig. 4), I remain unconvinced of the claim that the NuRD complex and V-ATPase specifically interact. Concerning the co-IP, the authors now mention that the co-IP was performed three times: "Assay was performed using three biological replicates. Three independent biological replicates of the experiment were conducted with similar results." However, the authors did not use ACT-4::GFP or GFP alone as controls for their co-IP as previously suggested. This is critical considering that the evidence for a specific HDA-1::GFP - V-ATPase interaction is rather weak (compare interactions between HDA-1::GFP and V-ATPase subunits in Fig 4B with those of HDA-1::GFP and subunits of NuRD in Fig S8B).

We conducted GFP pull-down experiments and MS spectrometric analysis forHDA-::GFP and ACT-4::GFP using identical protocols, yielding consistent results. We agree with the reviewer that in our Western blot, inclusion of ACT-4::GFP is amore effective negative control compared to empty beads.

(6) Based on Fig 5E, it appears that Bafilomycin treatment causes pleiotropic effects on animals (see differences in HDA-1::GFP signal in the three rows). The authors now state: "Although BafA1-mediated disruption of lysosomal pH homeostasis is recognized to elicit a wide array of intracellular abnormalities, we found no evidence of such pleiotropic effects at the organismal level with the dosage and duration of treatment employed in this study". However, the 'evidence' mentioned is not shown. It is critical that the authors provide this evidence.

We thank the Reviewer for pointing out this issue. We only checked the viability of the L1 larvae and morphology of animals at the organismal level with the BafA1 dosage and duration of treatment and did not notice any death of the animals and apparent abnormality in morphology (N > 20 for each treatment). However, as the reviewer pointed out, there can be some abnormalities at the cellular level. We thus revised this above description as the following, on page 11, line 27:

“…Although BafA1-mediated disruption of lysosomal pH homeostasis is recognized to elicit a wide array of intracellular abnormalities, we did not observe any larval deaths and apparent abnormality in morphology at the organismal level (N > 20 for each treatment) at the dose and duration of treatment employed in this study...”

The following is the authors’ response to the previous reviews.

**eLife assessment**
The authors propose that the asymmetric segregation of the NuRD complex in *C. elegans* is regulated in a V-ATPase-dependent manner, that this plays a crucial role in determining the differential expression of the apoptosis activator egl-1, and that it is therefore critical for the life/death fate decision in this species. If proven, the proposed model of the V-ATPase-NuRD-EGL-1-Apoptosis cascade would shed light onto the mechanisms underlying the regulation of apoptosis fate during asymmetric cell division, and stimulate further investigation into the intricate interplay between V-ATPase, NuRD, and epigenetic modifications. However, the strength of evidence for this is currently incomplete.
**Public Review:**
Xie et al. propose that the asymmetric segregation of the NuRD complex is regulated in a V-ATPase-dependent manner, and plays a crucial role in determining the differential expression of the apoptosis activator egl-1 and thus critical for the life/death fate decision.While the model is very intriguing, the reviewers raised concerns regarding the rigor of the method. One issue is with statistics (either insufficient information or inadequate use of statistics), and second is the concern that the asymmetry observed may be caused by one cell dying (resulting in protein degradation, RNA degradation etc). We recommend that the authors address these issues.

We extend our sincere thanks to the Editors and Reviewers for their insightful comments on this study.

Major #1:There are still many misleading statements/conclusions that are not rigorously tested or that are logically flawed. These issues must be thoroughly addressed for this manuscript to be solid.(1) Asymmetry detected by scRNA seq vs. imaging may not represent the same phenomenon, thus should not be discussed as two supporting pieces of evidence for the authors' model, and importantly each method has its own flaw. First, for scRNA seq, when cells become already egl-1 positive, those cells may be already dying, and thus NuRD complex's transcripts' asymmetry may not have any significance. The data presented in FigS1D, E show that there are lots of genes (6487 out of 8624) that are decreased in dying cells. Thus, it is not convincing to claim that NuRD asymmetry is regulated by differential RNA amount.

We agree with the reviewer's comment. Indeed, scRNA-seq reveals phenomena different from those observed in protein imaging, and NuRD asymmetry may not be regulated by differential RNA levels. Seven years ago, when we started this project, NuRD asymmetry during asymmetric neuroblast division was unknown. We first found NuRD mRNA asymmetry using scRNA-seq and then NuRD protein asymmetry using fluorescence imaging. We have documented the whole process of discovering NuRD asymmetry, although the asymmetry of NuRD complex transcripts does not necessarily imply protein asymmetry. We have revised statements related to "NuRD asymmetry being regulated by differential RNA amounts" and discussed this issue in the revised manuscript on page 14, line 2:

" The transcript asymmetry detected by scRNA-seq may not correspond to the protein asymmetry detected by microscopic imaging. Our scRNA-seq data shows that 6487 out of 8624 genes were not detected in egl-1-positive cells, the putative apoptotic cells. Cells that are egl-1 positive may be undergoing apoptosis, rendering the asymmetry of NuRD complex transcripts insignificant in inferring protein asymmetry. Thus, the observed transcript asymmetry of the NuRD subunits between live and dead cells may be coincidental with NuRD protein asymmetry during asymmetric neuroblast division, rather than serving as a regulatory mechanism."

(2) Regarding NuRD protein's asymmetry, there are still multiple issues. Most likely explanation of their asymmetry is purely daughter size asymmetry. Because one cell is much bigger than the other (3 times larger), NuRD components, which are not chromatin associated, would be inherited to the bigger cell 3 times more than the smaller daughter. Then, upon nuclear envelope reformation, NuRD components will enter the nucleus, and there will be 3 times more NuRD components in the bigger daughter cell. It is possible that this is actually the underling mechanism to regulate gene expression differentially, but this possibility is not properly acknowledged. Currently, the authors use chromatin associated protein (Mys-1) as 'symmetric control', but this is not necessarily a fair comparison. For NuRD asymmetry to be meaningful, an example of protein is needed that is non-chromatin associated in mitosis, distributed to daughter cells proportional to daughter cell size, and re-enter nucleus after nuclear envelope formation to show symmetric distribution. And if daughter size asymmetry is the cause of NuRD asymmetry, other lineages that do not undergo apoptosis but exhibit daughter size asymmetry would also show NuRD asymmetry. The authors should comment on this (if such examples exist, it is fine in that in those cell types, NuRD asymmetry may be used for differential gene expression, not necessarily to induce cell death, but such comparison provides the explanation for NuRD asymmetry, and puts the authors finding in a better context).

For more than one decade, we have meticulously explored the relationship between protein asymmetry and cell size asymmetry during ACDs of Q cells. A notable example of even protein distribution is the cytokinetic kinesin ZEN-4, as documented in our 2012 publication in the Journal of Cell Biology (Chai et al., JCB, 2012). This study, primarily focusing on the fate of the midbody post-cell division, also showcased the dynamics of GFP-tagged ZEN-4 during ACDs of QR.a cells in movie S1. Intriguingly, beyond its role in the cytokinetic ring, we observed a uniform dispersal of ZEN-4 throughout the cytoplasm. Remarkably, following cell division, ZEN-4 transitions evenly into the nuclei of the daughter cells, a phenomenon with implications yet to be fully understood. One hypothesis is that ZEN-4's nuclear localization may prevent the formation of ectopic microtubule bundles in the cytosol during interphase. Below, we present a snapshot from our original movie, clearly showing the symmetrical distribution of ZEN-4 into the nuclei of the two daughter cells.

(3) For the analysis of protein asymmetry between two daughters in Fig S4C, the method of calibration is unclear, making it difficult to interpret the results.

In Figure S4C, we quantified the relative total fluorescence of the Q cell, with the quantification method illustrated in Figure S4A. To further clarify our quantification approach, we have updated Figure S4A and the "Live-Cell Imaging and Quantification" section in the Materials and Methods:

“…To determine the ratios of fluorescence intensities in the posterior to anterior half (P/A) of Q.a lineages or A/P of Q.p lineages, the cell in the mean intensity projection was divided into posterior and anterior halves. ImageJ software was used to measure the mean fluorescence intensities of two halves with background subtraction. The slide background's mean fluorescence intensity was measured in a region devoid of worm bodies. The background-subtracted mean fluorescence intensities of the two halves were divided to calculate the ratio. The same procedure was used to determine the fluorescence intensity ratios between two daughter cells. Total fluorescence intensity was the sum of the posterior and anterior fluorescence intensities or the sum of fluorescence intensities from two daughter cells (Figure S4A). …”

(4) As for pHluorin experiments, the authors were asked to test the changes in fluorescence observed are due to changes in pH or changes in the amount of pHluorin protein. They need to add a ratio-metric method in this manuscript. A brief mention to Page 12 line 12 is insufficient to clarify this issue.

We appreciate the concerns about potential changes in pH or pHluorin protein levels. While we cannot completely dismiss the impact of changes in the amount of pHluorin protein, it appears improbable that the asymmetry of pHluorin fluorescence is attributed to an asymmetric amount of pHluorin protein. This inference is supported by the observation that other fluorescent proteins, such as GFP or mCherry, did not exhibit any asymmetry during ACDs of Q cells. An example of GFP alone during the ACD of QL.p is illustrated in figure 5A from Ou and Vale, JCB, 2009. The fluorescence intensities in the large QL.pa cell and the small QL.aa are indistinguishable.

Major #2:

Some issues surrounding statistics must be resolved.(1) Fig. 1FG, 2D, 3BDEG, 5BD and 6B used either one-sample t-test or unpaired two-tailed parametric t-test for statistical comparison. These t-tests require a verification of each sample fitting to a normal distribution. The authors need to describe a statistical test used to verify a normal distribution of each sample.(2) Fig. 2D, 3D, and 3G have very small sample size (N=3-4, N=6, N=3, respectively), it is possible that a normal distribution cannot be verified. How can the authors justify the use of one-sample t-test and unpaired parametric t-test ?(3) Statistical comparison in Fig. 2D and Fig. 6B should be re-assessed. For Fig. 2D, the authors need to compare the intensity ratio of HDA-1/LIN53 between sister cells dying within 35 min and those over 400 min. For Fig. 6B, they need to compare the intensity ratio of VHA-17 between DMSO- and BafA1- treated cells at the same time point after anaphase.

We appreciate the reviewer's advice on the statistical analysis of our data. In response, we performed normality tests on the datasets presented in Figures 1F, 1G, 3B, 5B, 5D, and 6B, all of which passed the tests (as demonstrated in Figure S10). We also acknowledge the reviewer's comment on the inadequate sample sizes in Figures 2D, 3D, 3E, and 3G for fitting a normal distribution. Therefore, we have revised our statistical analysis methods for these figures and updated both the figures and their legends. The revised statistical results support the primary conclusions of this study.

In response to the reviewer's observation regarding the small sample size in Figure 2D , which precluded normality verification, and the suggestion to compare sister cells that die within 35 minutes to those surviving over 400 minutes, we adapted our approach. We implemented the Kruskal-Wallis test to evaluate the differences among the groups. To assess the specific differences between each group and the 400 min MSpppaap group, we conducted the Dunn’s multiple comparisons test. The revised Figure 2D illustrates the updated statistical significance.

For Figure 3D, due to the small sample size precluding normality verification, we applied the Wilcoxon test with 1 as the theoretical median. The revised Figure 3D illustrates the updated statistical significance.

For Figure 3E, where the sample size also hindered normality verification, we conducted the Kruskal-Wallis test to evaluate the overall effect. Additionally, Dunn’s multiple comparisons test was utilized to examine the differences between groups. The revised Figure 3E illustrates the updated statistical significance.

For Figure 3G, the reviewer pointed out the small sample size and the limited statistical power due to having only three data points per group. To address this, we revised the figure to visually present each data point, aiming to more clearly illustrate the variation trends.

For Figure 6B, following the reviewer's suggestion, we compared the DMSO group directly with the Baf A1 group, updating Figure 6B to reflect this comparison as advised.

These adjustments have been made to ensure the statistical analyses are robust and appropriate given the sample sizes and to align with the reviewer's recommendations, enhancing the clarity and accuracy of our findings.

**Recommendations for the authors:**
We recommend using grey scale (instead of 'heatmap' representation) to show the protein distribution of interest. Heatmap does not help at all, because 'total protein amount per cell' (instead of signal intensity on each pixel) is what matters in the context of this paper. Heatmap presentation does not allow readers to integrate signal intensity with their eyes.

We thank the editor for pointing this out. We have changed heatmaps to inverted fluorescence images in grey scale.